METHODS AND RESOURCES

# DECODE enables high-throughput mapping of antibody epitopes at single amino acid resolution

Katsuhiko Matsumoto[1,2☯], Shoko Y. Harada[1☯¤], Shota Y. Yoshida[1,3☯], Ryohei Narumi[1], Tomoki T. Mitani[1,4,5], Saori Yada[2], Aya Sato[6], Eiichi Morii[3], Yoshihiro Shimizu[6], Hiroki R. Ueda[1,2,3,7] *

**1** Laboratory for Synthetic Biology, RIKEN Center for Biosystems Dynamics Research, Osaka, Japan, **2** Department of Systems Pharmacology, Graduate School of Medicine, The University of Tokyo, Tokyo, Japan, **3** Department of Pathology, Graduate School of Medicine, Osaka University, Osaka, Japan, **4** Department of Systems Biology, Graduate school of Medicine, Osaka University, Osaka, Japan, **5** Department of Neurology, Graduate school of Medicine, Osaka University, Osaka, Japan, **6** Laboratory for Cell-Free Protein Synthesis, RIKEN Center for Biosystems Dynamics Research, Osaka, Japan, **7** Institute of Life Science, Kurume University, Kurume, Japan

☯ These authors contributed equally to this work.
¤ Current address: Laboratory for Cell-Free Protein Synthesis, RIKEN Center for Biosystems Dynamics Research, Osaka, Japan
* uedah-tky@umin.ac.jp

**Data Availability Statement:** Underlying data is available in Zenodo: 10.5281/zenodo.14286317. This paper performed GPU-based epitope using a

## Abstract

Antibodies are extensively used in biomedical research, clinical fields, and disease treatment. However, to enhance the reproducibility and reliability of antibody-based experiments, it is crucial to have a detailed understanding of the antibody's target specificity and epitope. In this study, we developed a high-throughput and precise epitope analysis method, DECODE (Decoding Epitope Composition by Optimized-mRNA-display, Data analysis, and Expression sequencing). This method allowed identifying patterns of epitopes recognized by monoclonal or polyclonal antibodies at single amino acid resolution and predicted cross-reactivity against the entire protein database. By applying the obtained epitope information, it has become possible to develop a new 3D immunostaining method that increases the penetration of antibodies deep into tissues. Furthermore, to demonstrate the applicability of DECODE to more complex blood antibodies, we performed epitope analysis using serum antibodies from mice with experimental autoimmune encephalomyelitis (EAE). As a result, we were able to successfully identify an epitope that matched the sequence of the peptide inducing the disease model without relying on existing antigen information. These results demonstrate that DECODE can provide high-quality epitope information, improve the reproducibility of antibody-dependent experiments, diagnostics and therapeutics, and contribute to discover pathogenic epitopes from antibodies in the blood.

## Introduction

Antibodies play a crucial role in biological and biomedical research, as well as clinical applications like diagnostics and antibody-based therapies. Currently, there are over 7 million

genome-wide protein database in Figs 2 and S2. The codes for calculating DECODE score and visualization is available in Zenodo: 10.5281/zenodo.14221823.

**Funding:** This work was supported by a JST ERATO grant (H.R.U., no. JPMJER2001), a JST (Moonshot R&D) (K.M., no. JPMJMS2023), Science and Technology Platform Program for Advanced Biological Medicine JP21am0401011, AMED-CREST JP21gm0610006 (AMED/MEXT) (H. R.U.), a Brain/MINDS JP21dm0207049, Grant-in-Aid for Scientific Research (S) JP18H05270 (JSPS KAKENHI) (H.R.U.), a JSPS KAKENHI grant-in-aid for scientific research (c) (K.M., no. 20K06885), a JSPS KAKENHI grant-in-aid for Early-Career Scientists (S.Y. Y., no. 20K16626), a JSPS KAKENHI grant-in-aid for Early-Career Scientists (T. T. M., no. 20K16498), a grant-in-aid from the Human Frontier Science Program (H.R.U.), a MEXT Quantum Leap Flagship Program (MEXT QLEAP) (H.R.U., no. JPMXS0120330644) and an intramural Grant-in-Aid from the RIKEN BDR (H.R. U.).The funders had no role in study design, data collection and analysis, decision to publish, or preparation of the manuscript.

**Competing interests:** HR.U, KM, SY.H and YS have cpatents filed for DECODE. HR.U and KM are concurrently working for CUBICStars Inc.

**Abbreviations:** CFA, complete Freund's adjuvant; DECODE, Decoding Epitope Composition by Optimized-mRNA-display, Data analysis, and Expression sequencing; EAE, experimental autoimmune encephalomyelitis; ELISA, enzyme-linked immunosorbent assay; GCE, genetic code expansion; HRP, horseradish peroxidase; MOG, myelin oligodendrocyte glycoprotein; NGS, next-generation sequencing; NMR, nuclear magnetic resonance; PTM, posttranslational modification.

antibodies listed worldwide [1]. Despite the acceleration in antibody production, the reproducibility and reliability of antibody-based studies have been a longstanding concern [2–5]. These issues often stem from antibody quality, including factors like purity, affinity, specificity, and cross-reactivity. While purity and affinity can be quantitatively assessed using indicators like titer and $K_d$, there is a lack of sufficient indicators for specificity and cross-reactivity. Epitope information, which refers to the site an antibody recognizes, is valuable for evaluating antibody specificity and cross-reactivity. Typically, antibodies recognize 10 or fewer amino acid residues when binding to linear epitope, with the most critical being the five or fewer hotspot residues energetically required for binding [6,7]. Understanding the significance and characteristics of hotspot residues in epitopes can assist researchers in selecting antibodies that are most suitable for their experimental conditions or in designing the optimal conditions for these antibodies. Additionally, by exploring sites similar to the epitope across genome-wide protein sequences, it becomes possible to predict antibody target specificity and cross-reactivity with non-targets. Leveraging such detail epitope information has the potential to enhance the reproducibility of antibody-dependent scientific investigations and pathological diagnoses. Despite the significance of detailed epitope information, commercially available antibodies remain largely uncharacterized in this regard. Epitope databases like IEDB and IMGT, as well as antibody manufacturer websites, rarely provide such details. Addressing this challenge necessitates the development of high-throughput genome-wide epitope analysis methods with single amino acid resolution.

Many epitope analysis methods have been reported to date, but achieving high-resolution, comprehensiveness, and genome-wide analysis simultaneously has been challenging due to their limitations in detection sensitivity and throughput [8–15]. Although HDX-MS, X-ray crystallography and nuclear magnetic resonance (NMR) methods allow for 3D structure observation, they require an antigen itself and have low throughput [9–11]. Epitope mapping using peptide libraries, such as peptide microarrays [12] and peptide selection [13–19], is effective for identifying binding sites and hot spot residues. The theoretical diversity that hotspot residues in an epitope can be calculated to approximately $10^9$ by raising the number of hotspot residues to the 20th power and then multiplying by the number of possible positions of the hotspot residues in the binding site. In other words, to accurately identify hotspot residues, a library size that exceeds this diversity is required. Peptide selection by peptide microarrays [12] or bacterial display [13] limits the range or quality of epitope searches due to the small size of peptide libraries (approximately $<10^5$ and $<10^6$, respectively). Among peptide selection methods, the library size of phage display [14–16] is approximately $<10^9$, and the library size of mRNA display [17,18] and of ribosome display [19] are approximately $<10^{13}$. Given that these methods can handle large libraries, they are anticipated to accurately identify patterns of significance in the amino acids recognized by antibodies within epitopes, including hotspot residues. In particular, mRNA display and ribosome display are in vitro translation-based peptide selection methods, where all components are well defined and controllable, providing scope for developing simpler, higher-throughput protocols. Additionally, mRNA display methods have the advantage of eliminating large ribosomes that may interfere with peptides binding to the target molecule due to the formation of covalent bonds between the mRNA and the translation product via puromycin. However, conventional mRNA display protocols are complex and time-consuming, making it difficult to apply them as tools for analyzing epitopes of large numbers of antibodies.

In this study, we developed a simple, efficient, and high-throughput epitope analysis method called DECODE (Decoding Epitope Composition by Optimized-mRNA-display, Data analysis, and Expression sequencing). This method enables comprehensive and detailed epitope analysis with single amino acid resolution for antibodies that recognize linear epitopes

(Fig 1A). Using multiwell plates, we thoroughly identified the sites and significant amino acid patterns recognized by monoclonal, polyclonal, and serum antibodies, and predicted their cross-reactivity across the genome. Enzyme-linked immunosorbent assay (ELISA) experiments demonstrated that the antibodies precisely bound to the identified epitope at the single amino acid level. The detailed epitope information obtained from DECODE not only aided in selecting appropriate antibodies to improve the reproducibility of immunostaining formalin-fixed samples but also contributed to the development of novel 3D immunostaining methods, enhancing the permeability of antibodies. Furthermore, DECODE re-identified pathogenic epitopes from serum-derived antibodies in autoimmune disease models without prior antigen information. Here, we demonstrate that DECODE can enhance the scientific reliability and reproducibility of experiments that rely on antibodies. Furthermore, we show the potential of DECODE to detect pathogen-specific epitopes from antibodies in blood, even when information about the antigen is unknown.

## Results

### The development of the high-throughput and detailed epitope analysis method, DECODE

To achieve high-throughput and detailed epitope analysis, we developed the DECODE method, which consists of an improved mRNA display method and GPU-based next-generation sequencing (NGS) analysis. This chapter provides details about the optimized mRNA display method. Conventional mRNA display methods have low yields of products from each reaction step, including transcription, ligation, and translation so that these methods require multiple purification steps to remove unreacted regents and change to the appropriate reaction buffer, which hinder improvements in throughput [17,18,20]. We improved the buffer composition and reaction conditions for all these steps and optimized them to improve product yields as shown below.

Initially, a template DNA library was designed using a random sequence of 12 amino acids to sufficiently accommodate the nearly linear epitope length and using NNK to reduce the frequency of stop codons (S1A Fig). Our significant improvement was the enhanced ligation efficiency between mRNA and puromycin-conjugated DNA (pu-DNA). With conventional methods, this ligation efficiency was low, requiring electrophoresis and purification, which we considered to be an obstacle to achieving high-throughput selection and a cause of decreased library yield. We investigated the cause of the low ligation efficiency between mRNA and pu-DNA. We confirmed the ligation between mRNA and pu-DNA using polyacrylamide gel electrophoresis with single-nucleotide resolution and found that approximately half of the mRNA had one base added during the transcription reaction, which interfered with ligation (S1B Fig). Previous papers have shown that base addition occurs due to run-off of T7 RNA polymerase [21]. To reduce base addition by T7 RNA polymerase, we used template DNA with two $2'$-O-methyl-guanosine (Gm) residues introduced at the $5'$ end of the complementary strand. As a result, the transcript ligation efficiency increased from 53% to 91% (Figs 1B, S1B, and S1C).

Next, we examined the ligation efficiency of mRNA and peptide depending on the type of ligation approach. When we compared the linear ligation approach with the hairpin-format ligation approach, we found that linear ligation using Gm-containing templates had the highest ligation efficiency (S1D–S1F Fig). Regarding the recovery of peptide selection, linear ligation also showed higher recovery than hairpin ligation (S1G Fig). Additionally, we were concerned about the degradation of the mRNA-peptide complex and used the RNase- and Protease-free PURE system for in vitro translation [22] and performed the binding reaction to

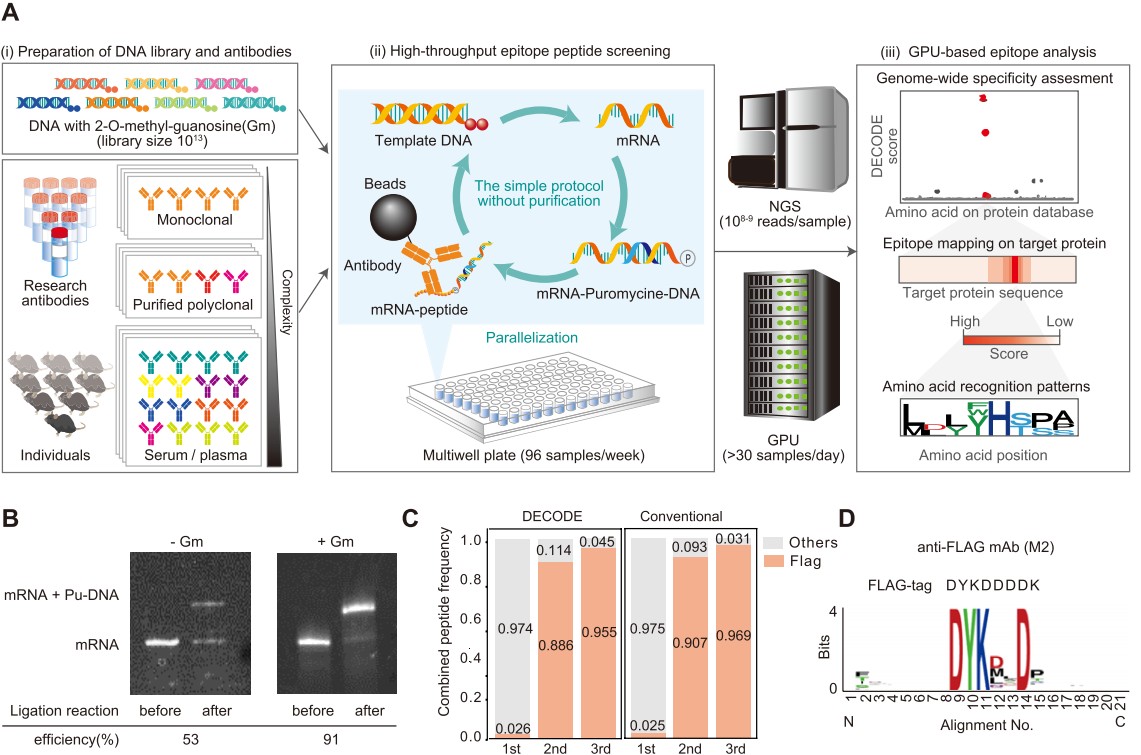

**Fig 1. Overview of the high-throughput and deep epitope analysis named DECODE.** (A) Overview of epitope analysis by DECODE. (i) DECODE is based on a $10^{13}$ DNA library containing two 2′-O-methyl-guanosine (Gm) in the 5′ end of the antisense strand. Target antibodies can be applied not only to monoclonal but also to complex polyclonal or serum antibodies. (ii) To increase the parallelization, the DECODE peptide selection method was constructed by reducing the purification process of mRNA display, and it achieved peptide selection using multiwell plates in a week. (iii) GPU-based epitope analysis against the genome-wide protein database. Enriched peptide sequences are read out by NGS and identify the epitope information in a single amino acid resolution such as target protein, binding position, and recognized residues by antibodies. (B) Comparison of ligation efficiencies of mRNA to puromycin-conjugated DNA (Pu-DNA) associated with transcription products derived from DNA template without (left panel) and with (right panel) Gm modification. Before and after ligated products were separated and quantified by polyacrylamide gel electrophoresis and SYBR Gold staining. (C) Laminated bar graph showing the ratios of Flag motif in recovered peptides on each round of DECODE selection or conventional mRNA display. Red and gray indicate ratios of the Flag motif and others, respectively. DECODE, mean: 1st: 0.026, 2nd: 0.886, 3rd: 0.955, STD: 1st: 0.011, 2nd: 0.019, 3rd: 0.013, $n = 4$. Conventional, mean: 1st: 0.025, 2nd: 0.907, 3rd: 0.969, STD: 1st: 0.002, 2nd: 0.014, 3rd: 0.003, $n = 4$. (D) Sequence logo of anti-Flag monoclonal antibody (clone M2) derived from the most convergent top 1,000 peptides in the third round of DECODE selection. This logo was generated using WebLogo. The upper sequence indicates the FLAG-tag sequence. The data underlying for panels C and D shown in the figure can be found in S1 Data or https://doi.org/10.5281/zenodo.14286317. DECODE, Decoding Epitope Composition by Optimized-mRNA-display, Data analysis, and Expression sequencing; NGS, next-generation sequencing.

targets before reverse transcription. As described above, we succeeded in omitting the electrophoresis and purification steps by reducing the amounts of unreacted products in each step.

To demonstrate this optimized protocol for epitope analysis, peptide selection was performed to the anti-FLAG antibody (clone M2) as a target. It has already been reported that the epitope motif recognized by the anti-FLAG antibody is DYKXXD [20,23]. The NGS results showed that the proportion of the known epitope motif DYKXXD increased through each round of peptide selection by the DECODE method, reaching over 95% in the third round (Fig 1C). This result indicates that the peptide enrichment efficiency of the DECODE method is almost the same as that of conventional methods, which require the gel electrophoresis purification of mRNA and Pu-DNA complexes. This demonstrates that the DECODE method can achieve higher throughput without the laborious electrophoresis and purification step, while

maintaining peptide enrichment efficiency. Sequence logos were retrieved from the top 1,000 peptide sequences of the NGS data by WebLogo [24–26] (Fig 1D). These results indicated that the new improved mRNA display method can sufficiently recover the epitope peptides of the antibody. This protocol is designed so that the reaction solution composition at each step does not affect the next reaction, allowing the process to proceed with simple pipetting. Ultimately, it will be possible to perform peptide selection targeting at least approximately 100 individual antibodies per week using multiwell plates, increasing the throughput by more than 100 times using existing methods.

## GPU-based epitope analysis using a genome-wide protein database

To predict the specificity and genome-wide cross-reactivity of a large number of antibodies from the NGS results obtained by DECODE, we developed a GPU-based analysis algorithm with reference to previous study [27]. First, we used the BLOSUM62 table, which is a homology score matrix between amino acids commonly used to compare amino acid sequences, for quantitative evaluation of sequence similarity, and the negative values were modified to zero because DECODE selection used the randomized library (S2L Fig). Next, we calculated a frequency list of peptides which had $20^{12}$ random amino acids generated from an ideally random cDNA library ($4^{36}$ bases) and derived sequence similarities (similarity scores) between this list of peptides and all positions of the protein referenced from the UniProt database (S2A Fig). The theoretical inverse cumulative probability distribution (P) map was prepared from the similarity scores and the peptide frequency list (Fig 2A, top). The experimental inverse cumulative probability distribution (Q) maps were similarly calculated from the cDNA sequences of the DECODE selections with 100,000 to 1 million reads from the NGS results (Fig 2A, bottom and S2B Fig). Fig 2 shows the analysis results of the anti-c-Fos antibody (clone 9F6) as an example. In comparison with the P-map, the similarity scores around the binding site of Q-map were increased (Fig 2B and 2C). To quantify this difference, we explored various formulas commonly used in machine learning to calculate the distances between various distributions. As a result, the Pearson X2 formula showed the highest distance around the high similarity score area, while the low similarity area showed the smallest distance (Figs 2D and S2C). These results indicated that the presence of more similar sequences could be detected with high sensitivity. The sum of the distances of similarity score for each protein position between the P and the Q was named the DECODE score. The DECODE scores at all positions of the c-Fos protein (Fig 2E) showed a remarkably high peak around the binding site. We also plotted the DECODE scores of all mouse proteins (Fig 2F), and the highest DECODE score was found for the c-Fos protein. The top 100 DECODE scores are plotted in Fig 2G, which shows that only a portion of the c-Fos protein had a high DECODE score. These results suggest that clone 9F6 may have high target specificity. Other clones of the c-Fos antibody were also analyzed, and the DECODE scores of the c-Fos protein were visualized, showing a peak for each antibody (Figs 2H and S2D–S2H). DECODE score calculations were performed on the GPU, which was coded using CUDA C/C++ to perform the calculations faster for multiple NGS results, and 1 million lead NGS results were achieved in 3 min. DECODE scores were also calculated using the WAC table, another amino acid similarity score table, which has a maximum value of 4 and a narrower range of similarity scores that enabled a reduction in computation time, and similar results were obtained (S2I–S2K and S2M Fig). The WAC table could be used as an alternative to BLOSUM62 if there is a need to reduce the computational time. As above, this analysis method has enabled a comprehensive estimation of the protein, site, and amino acid to which an antibody binds among all proteins across the genome.

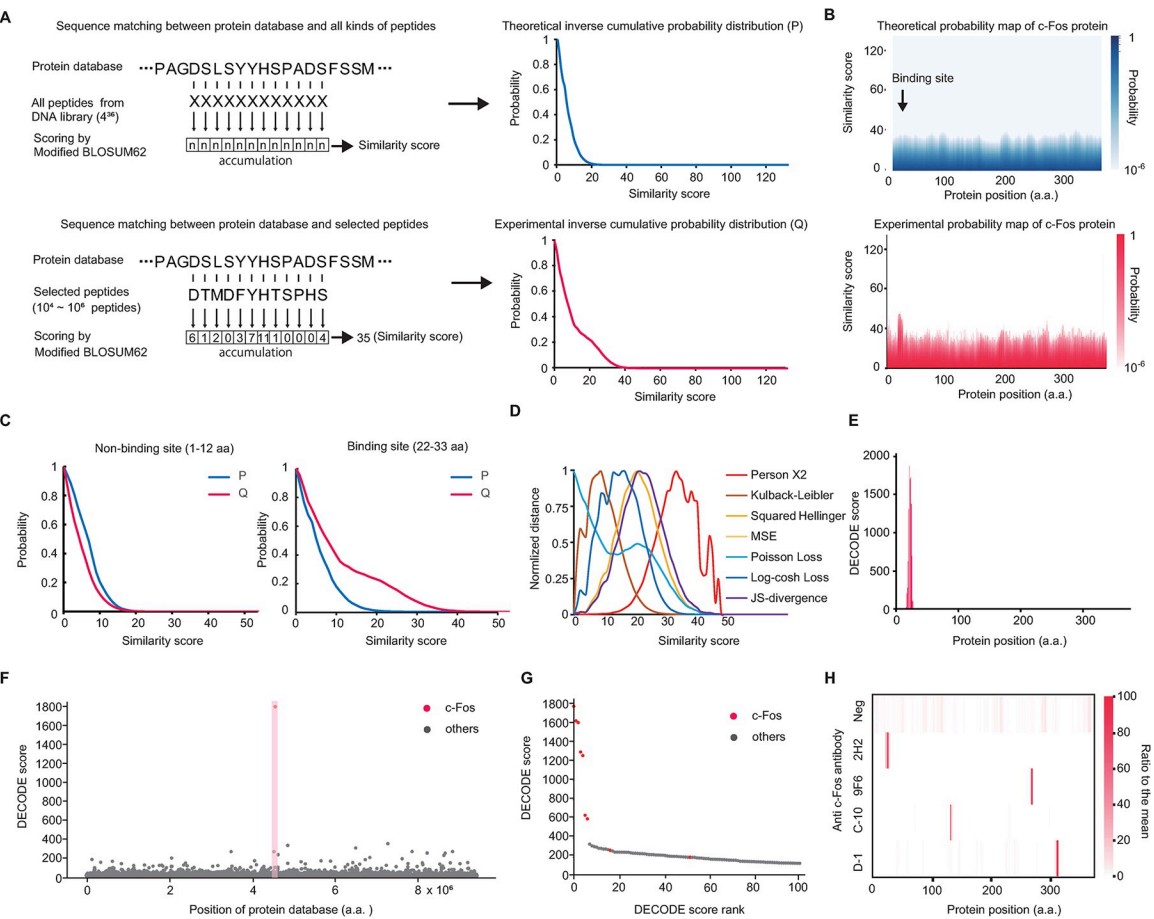

**Fig 2. A GPU-based deep epitope analysis in DECODE using the genome-wide protein database.** (A) Calculation flows of the theoretical inverse cumulative probability distribution (P) between the protein database and all possible peptides from the random DNA library (upper) and experimental inverse cumulative probability distribution (Q) between the protein database and screened peptides by DECODE selection (lower). (B) Heat map of the inverse cumulative probability distribution of each position of the mouse c-fos protein. Theoretical distribution (upper) or experimental distribution by DECODE selection for anti-c-fos antibody clone 9F6 (bottom). (C) Inverse cumulative probability distribution plots on the non-binding site (left) and the binding site (right) of anti-c-fos antibody clone 9F6. (D) Plots of the distances at each similarity score between P and Q on the binding site are calculated by various distance functions. (E) A bar graph of the DECODE score on the mouse c-fos protein sequence for anti-c-fos antibody clone 9F6. (F) Manhattan plot of all mouse proteins for anti-c-fos antibody clone 9F6. Red dots and highlights indicate the c-fos protein. This plot was visualized with downsampled data to 1/50 using LTTB [51]. (G) The top 100 DECODE score showed in (F). Red dots indicate the c-fos protein. (H) Epitopes map of 4 monoclonal anti-c-fos antibodies against the c-fos protein. These DECODE scores were normalized by the ratio with the means on c-fos protein. The data underlying for panels A–H shown in the figure can be found in S1 Data or https://doi.org/10.5281/zenodo.14286317. DECODE, Decoding Epitope Composition by Optimized-mRNA-display, Data analysis, and Expression sequencing; LTTB, Largest Triangle Three Buckets.

## DECODE identified the binding sites and hotspot residues

Epitope analysis of anti-c-Fos antibodies (clones 2H2, 9F6, and C-10) was conducted using the DECODE method to identify the sites where antibodies bind and hotspot residues (Fig 3A). Sequence logos were visualized only for amino acids that converged by 20% or more. For all antibodies, the same epitope was reproducibly obtained in 3 independent peptide selection (S3A Fig). The analysis results also showed that each antibody specifically recognized a different site on the c-fos protein (S3B Fig), and there was no correlation between clones (S3C Fig). We performed the ELISA to verify whether the predicted epitopes are actually specifically bound by the 3 anti-c-Fos antibodies. First, we chemically synthesized the identified epitope

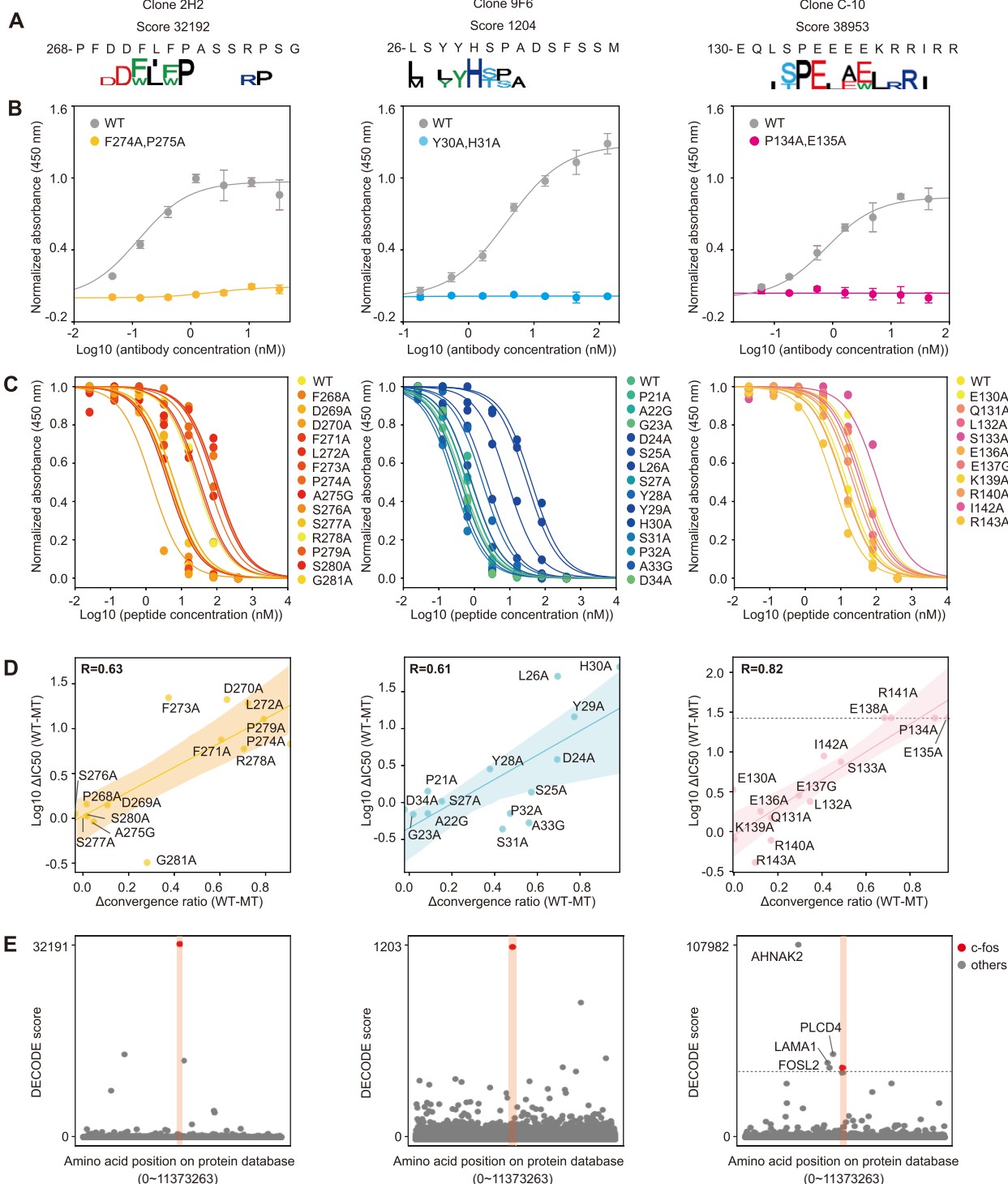

**Fig 3. DECODE identifies the binding sites and hot spot residues.** (A) Sequence logos at the highest DECODE score position at round 3 of the DECODE selection for anit-c-fos monoclonal antibodies (9F6, 2H2, and C-10). (B) Binding assay for anti-c-fos antibodies (clones 2H2, 9F6, and C-10) at each concentration against mouse c-fos protein by direct ELISAs. There was immobilized wild-type protein (gray) or double mutated proteins (light blue, yellow, and pink), respectively. The A450 was normalized by the saturated value of antibodies with different epitopes provided as S3F–S3H Fig. Data are shown as means ± STD ($n$ = 3). Lines represent fitting with Michaelis–Menten equation. (C) Competitive assay for anti-c-fos antibodies (clones 9F6, 2H2, and C-10) to mouse c-fos protein with every single amino acid mutated peptide at various concentrations. A450 values were normalized by non-competitive conditions shown in S4A Fig. Data are shown as means ± STD ($n$ = 3). Lines represent fitting with Michaelis–Menten equation. Mutant peptide sequences are provided as S4 Table. (D) Scatter plot of the correlation between the difference in

converged amino acid ratios between wild and mutant amino acids (Δconvergence ratio (WT-MT)) and the difference of IC50 calculated in (C) between mutant and wild type (Log10 ΔIC50 (WT-MT)). Plots, lines, and shaded areas represent actual data, the regression line, and the 95% confidence bounds, respectively. Spearman correlation coefficients about 3 clones 9F6, 2H2, and C-10 were respectively R = 0.61, R = 0.63, and R = 0.82. (E) Manhattan plots of the DECODE scores on the human protein database for each anti-c-fos antibody (clones 9F6, 2H2, and C-10). Plots for the c-fos protein are shown in red. These plots visualize data downsampled to 1/50 using LTTB. The data underlying for panels A–E shown in the figure can be found in S1 Data or https://doi.org/10.5281/zenodo.14286317. DECODE, Decoding Epitope Composition by Optimized-mRNA-display, Data analysis, and Expression sequencing; ELISA, enzyme-linked immunosorbent assay; LTTB, Largest Triangle Three Buckets.

peptides, which were the 12 consecutive amino acid sequence that had the highest DECODE score in the target protein sequences including several residues before and after, and investigated their binding reactions with each antibody. The results showed that each antibody specifically recognized the sequences identified by DECODE (S3D Fig). We next performed ELISA using a double mutant c-Fos protein with 2 predicted hotspot residues mutated (Figs 3B, S3E and S3F). As a result, each antibody bound to the wild type but not to the mutant c-Fos, suggesting that the sites where antibodies bind was accurately predicted by DECODE. Also, when antibodies with different epitopes were used, there was a binding reaction for both the wild type and the mutant (S3G and S3H Fig). These results supported that each antibody actually binds specifically to the site identified by DECODE. Furthermore, to verify the accuracy of hotspot residues identified by DECODE, we performed competitive ELISA against c-Fos protein using epitope peptides with single amino acid mutations (Figs 3C, S4A and S4B). We confirmed a correlation between the amount of change in DECODE score due to mutations and the IC50 calculated as a result of competitive ELISA (Fig 3D). These correlations suggested that peptides with mutations in the hotspot residue significantly bind to antibodies less effectively than peptides with the other mutations. The above results mean that DECODE identify accurate epitopes at the single amino acid level and quantify variations in recognition by antibodies. Finally, to estimate genome-wide cross-reactivity for the 3 antibodies, we calculated DECODE scores for all mouse proteins derived from UniProt (Fig 3E). The results showed that clones 2H2 and 9F6 had high specificity for the c-fos protein, whereas C-10 showed cross-reactivity with other proteins. Among the proteins predicted to cross-react with C-10, we particularly focused on 4 proteins (AHNAK2, LAMA1, FOSL2, and PLCD4) that had higher DECODE scores than the target protein (c-fos) (Fig 3E). There were observed that the sites with the maximum DECODE scores on these proteins contain sequences very similar to the motif of C-10. In particular, FOSL2 belongs to the same FOS family as the c-fos protein, and its sequence is very similar, with a close DECODE score value. The ELISA results for these sequences showed that anti-c-fos (C-10) cross-reacts with AHNAK2, LAMA1, and FOSL2 (S3I Fig). The antigen peptide derived from PLCD4 (pep4) was not soluble and could not be tested in ELISA. These results indicate that in immunohistochemistry, C-10 may stain not only the target protein c-fos, a neuronal activity marker, but also other proteins, leading to a risk of misinterpretation of research results. This highlights the importance of predicting cross-reactivity in immunohistochemical research.

Therefore, it can be said that the DECODE results may serve as a standard for evaluating the quality of antibodies, including their target specificity. The above series of experiments with the anti-NeuN antibody clone A60, which targets a different protein, NeuN, demonstrated consistency of DECODE (S4C–S4G Fig).

## DECODE revealed epitope similarities and differences

DECODE was performed on a total of 230 antibodies, and several remarkable findings were obtained (S1 Table). The peptide selection for these antibodies was conducted in 3 rounds. The cDNA-peptide complexes obtained in each round were quantified by qPCR, and it was

found that sufficient peptides had converged by the third round. Focusing on the epitopes of the anti-p53 monoclonal antibodies, we found similar epitopes despite their different clones, between clones Bp53-12, DO-1, and DO-7, and also between C-11 and D-11 (Figs 4A, 4C, S5A and S5B). First, results of ELISA indicated that 8 kinds of anti-p53 antibodies specifically bound to each individual epitope peptide and p53 protein (Figs 4D and S6A). Competitive ELISA using single amino acid mutant epitope peptides for clones Bp53-12, DO-1, and DO-7, which showed strong affinity, ensured the accuracy of the identified epitope hotspot residues (Figs 4B and S5C–S5E). These 3 clones also showed similar Km values to p53 protein (Figs 4E and S6B). As described above, both epitope analysis results and ELISA results showed that Bp53-12, DO-1, and DO-7 recognize the same binding site at the single amino acid level. Therefore, we next investigated whether the antibodies in these 3 clones with different name were actually the same or different. We performed quantitative LC-MS analysis to compare the sequences between the proteins for Bp53-12 and DO-1, which are both mouse IgG2a kappa proteins, to determine whether they were the same antibody or not. The antibodies and the internal control proteins, lysozyme, and cytochrome C, were digested with trypsin, and the resulting peptides were labeled with light (28 Da) and heavy (32 Da) isotopes containing dimethyl groups to identify the source proteins. As a result, the signal intensities of the peptides from the same antibody and internal control were almost the same for both the light and the heavy fragment; however, the variable region-derived peptides of clone Bp53-12 and DO-1 showed distinctive signals, respectively (Fig 4F). DO-7 (mouse IgG2b kappa) and Pab240 (mouse IgG1 kappa) were also compared with Bp53-12 by LC-MS and were confirmed to contain different peptides (S6C and S6D Fig). These results indicate that antibodies with different variable regions and isotypes may recognize nearly identical amino acids, a finding that can be readily revealed by DECODE.

Polyclonal antibodies are still used in biological research, although it is widely perceived that there is poor reproducibility among lots. We analyzed 2 different products of anti-c-Fos polyclonal antibodies' products, which respectively had 2 lots made from different individuals: cat#ab190289 (lot# GR3253255-1, GR3313102-1, Abcam) and cat#ab209794 (lot# GR3198011-8, GR3266315-7, Abcam). We confirmed the binding between these antibodies and 15 types of c-Fos protein fragment peptides identified by DECODE (Figs 4G and S7A–S7C). As an ELISA result, we found that despite having the same product number, both lots bound to almost different positions on the target and the correlation between lots was low (Figs 4H and S7D).

## DECODE's epitope information helps to optimize immunostaining fixation conditions and improves antibody permeation rates in 3D immunostaining

We attempted to apply the epitope information from DECODE to 3D immunostaining. First, we demonstrated the prediction of the sensitivity of formalin/PFA fixation in immunostaining based on the epitope information for 2 anti-c-Fos monoclonal antibodies, 2H2 and 9F6. According to DECODE results, clone 9F6 recognized the hotspot residues contained Tyr and His, which form irreversible methylene bridges via primary amines and formalin/PFA [28], that is why it was predicted to decrease the signals for strong fixed samples (S8A Fig). We certified the correctness of these predictions by ELISA (S8B Fig), and 3D immunostaining using normally fixed and strongly fixed mouse whole brains (Figs S8C–S8F and S9). As a result, 9F6 decreased the signals than 2H2 due to the influence of fixation, and antigen retrieval was also ineffective. Thus, it is possible to select an appropriate antibody for each formalin fixation condition based on the detailed epitope information obtained by DECODE. This is expected to enhance the reproducibility of experiments.

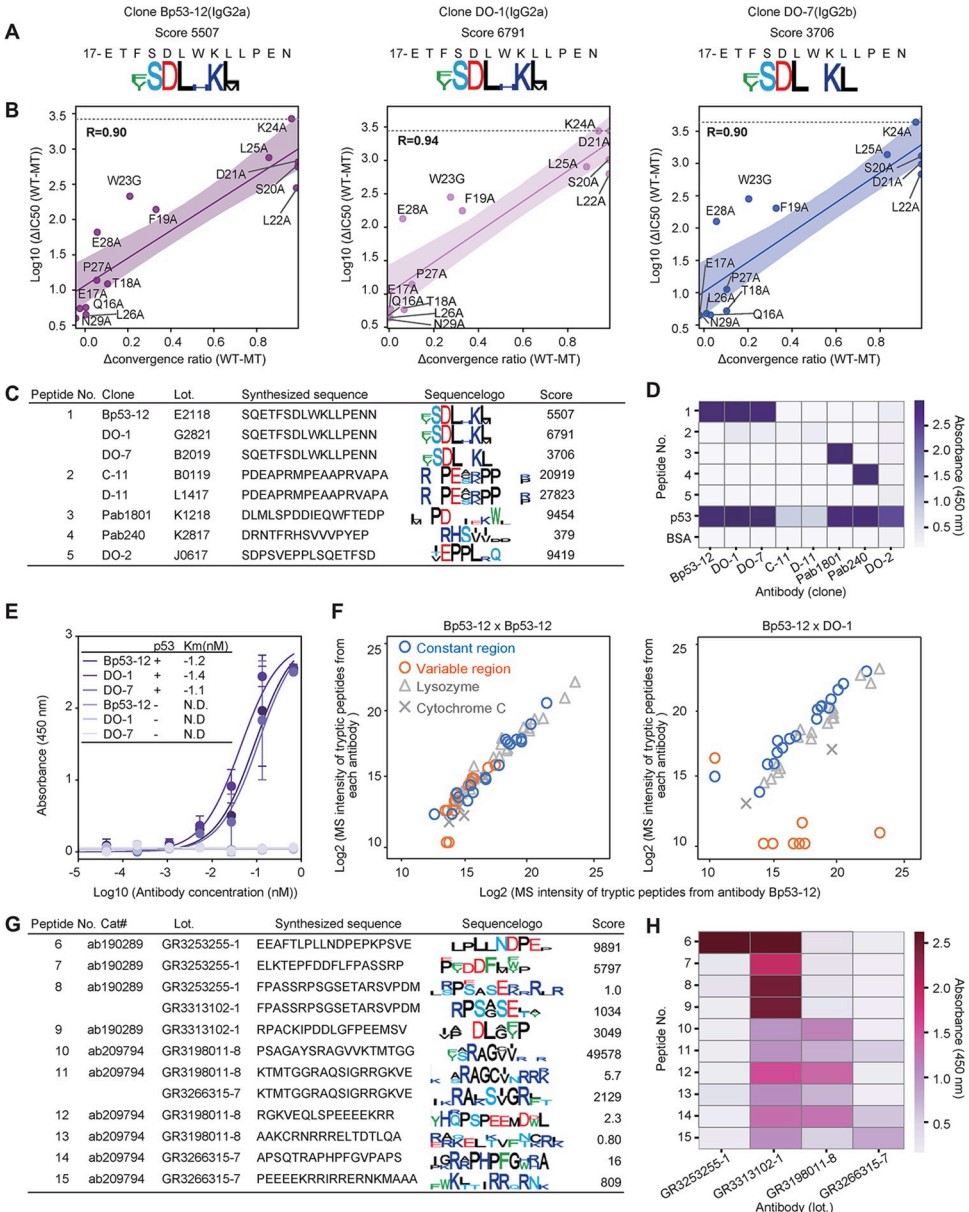

**Fig 4. DECODE reveals epitope similarities among monoclonal antibodies and differences among distinct clones for polyclonal antibodies.** (A) Epitope sequence logos of anti-p53 monoclonal antibodies (Bp53-12, DO-1, and DO-7) at third round of DECODE selection. (B) Correlation between the difference in convergent amino acid ratios between wild and mutant amino acids (Δconvergence ratio (WT-MT)) and the difference of IC50 calculated in S4C Fig between mutant and wild type (Log10 ΔIC50 (WT-MT)). Lines and shaded areas represent the regression line and the 95% confidence limits, respectively. Spearman correlation coefficients for 3 clones (Bp53-12, DO-1, and DO-7) were respectively R = 0.90, R = 0.94, and R = 0.90. (C) Synthesized peptides for binding assays and epitope logos for each anti-p53 monoclonal antibody. (D) Results of the direct ELISAs for each anti-p53 antibody against each peptide shown in (C), p53 protein, and BSA. Data are shown as means ± STD (*n* = 3). (E) Binding curves of direct ELISAs and Km values for anti-p53 antibodies (clones Bp53-12, DO-1, DO-7). Data are shown as means ± STD (*n* = 3). Lines represent fitting with Michaelis–Menten equation. (F) Correlation of the signal intensities of LC-MS during anti-p53 antibodies. Antibodies were digested by trypsin and their amino groups were labeled by dimethyl group with isotope reagents to identify each antibody. Blue circles and orange circles indicate the constant and variable regions in each antibody. The gray triangles and the cross show the peptides of lysozyme and cytochrome C proteins, respectively, which were added to each sample as internal standards. (G) The catalog number, lot number, synthesized peptides for binding assays, and epitope logos for anti-c-Fos polyclonal antibodies. (H) Binding assay with direct ELISAs for each antibody against each peptide shown in (G). Data are shown as means (*n* = 3). The data underlying for panels A–H shown in the figure can be found in S1 Data or https://doi.org/10.5281/zenodo.14286317. DECODE, Decoding Epitope Composition by Optimized-mRNA-display, Data analysis, and Expression sequencing; ELISA, enzyme-linked immunosorbent assay.

Furthermore, by applying epitope information, we can address the issue of antibody permeability, which is a significant challenge in 3D immunostaining. Generally, when performing 3D immunostaining on a cleared whole mouse brain or large organ without slicing, the antibodies tend to be trapped on the tissue surface and do not penetrate into the center, making uniform immunostaining difficult. To investigate the important factors for the penetration of the antibody into large organs, referring to previous studies [29], we simulated the concentration of the antigen-antibody complexes using various parameters: antibody concentration, antigen concentration, diffusion rate, and staining duration (Figs 5A and S10A–S10D). The results suggested that the low $k_{on}$ improved the penetration speed, but the complex concentration was decreased (i.e., the signal). However, in many cases, low antibody permeability is due to the antigen concentration being too high, which means that even if permeability is increased, sufficient signal intensity can be expected for observation. Therefore, we hypothesized that a strategy to improve antibody permeability by reducing the apparent $k_{on}$ of antibodies using chemically synthesized epitope peptides might be effective for 3D immunostaining. We first obtained the epitope information of anti-NeuN and anti-tyrosine hydroxylase (TH) monoclonal antibodies, which are difficult to stain in the whole mouse brain, and the accuracy of these epitopes was demonstrated by competitive ELISA (Figs 5B–5D and 5G–5I). We next immunostained NeuN or TH proteins in whole mouse brains with each antibody, both in the presence and absence of competitive epitope peptides that reduces the apparent $k_{on}$. To compare the antibody penetration between with and without epitope peptides, we aligned brain samples to a representative sample as a fixed image using ANTs software and autofluorescence images (S10E and S10F Fig) [30,31]. The results indicated that the permeabilities of these antibodies increased in the cerebellum and striatum, regions typically difficult for antibody penetration, when a competing peptide was added. Interestingly, when using the peptides with double mutations in hotspot residues, the penetration speed was not increased (Figs 5E, 5F, 5J–5M and S11). Furthermore, we performed co-staining for the NeuN and TH proteins using the optimal concentration of the full-match peptide. The penetration speed of each antibody was compared in the cross-sections and was confirmed to be improved (Fig 5N and 5O). These results support the hypothesis that epitope peptides identified by DECODE enhance the permeability of 3D immunostaining by individually controlling the apparent $k_{on}$ of each antibody. This highly antibody-specific strategy demonstrated that co-staining can also be readily achieved.

## DECODE re-identified the pathogenic epitope from serum antibodies in experimental autoimmune encephalomyelitis (EAE) mice

To demonstrated the applicability of DECODE to more complex blood antibodies, we performed epitope analysis of serum antibodies of EAE mouse. Assuming that the cause of the disease is unknown, here we attempted to re-identify a sequence that matched a peptide to induce the disease without relying on prior antigen information, methods such as constructing a library based on the antigen protein sequence, purifying specific antibodies using the antigen protein, or extracting peptides are not utilized. Initially, we confirmed the sensitivity of DECODE against a highly complex antibody mixture by titrating the anti-FLAG antibody, which mimics the same antigen's antibody group in healthy mouse serum. As a result, the FLAG motif was detected at antibody concentrations exceeding 1.1% and 0.28% in the second and third rounds, respectively (Figs 6B and S12A–S12C). Subsequently, we prepared EAE mice by injecting the myelin oligodendrocyte glycoprotein (MOG) fragment peptide (MOG35-55) with complete Freund's adjuvant (CFA). We immobilized IgG antibodies from EAE mouse serum with a score of 3 or non-immunized healthy mice on protein G magnetic

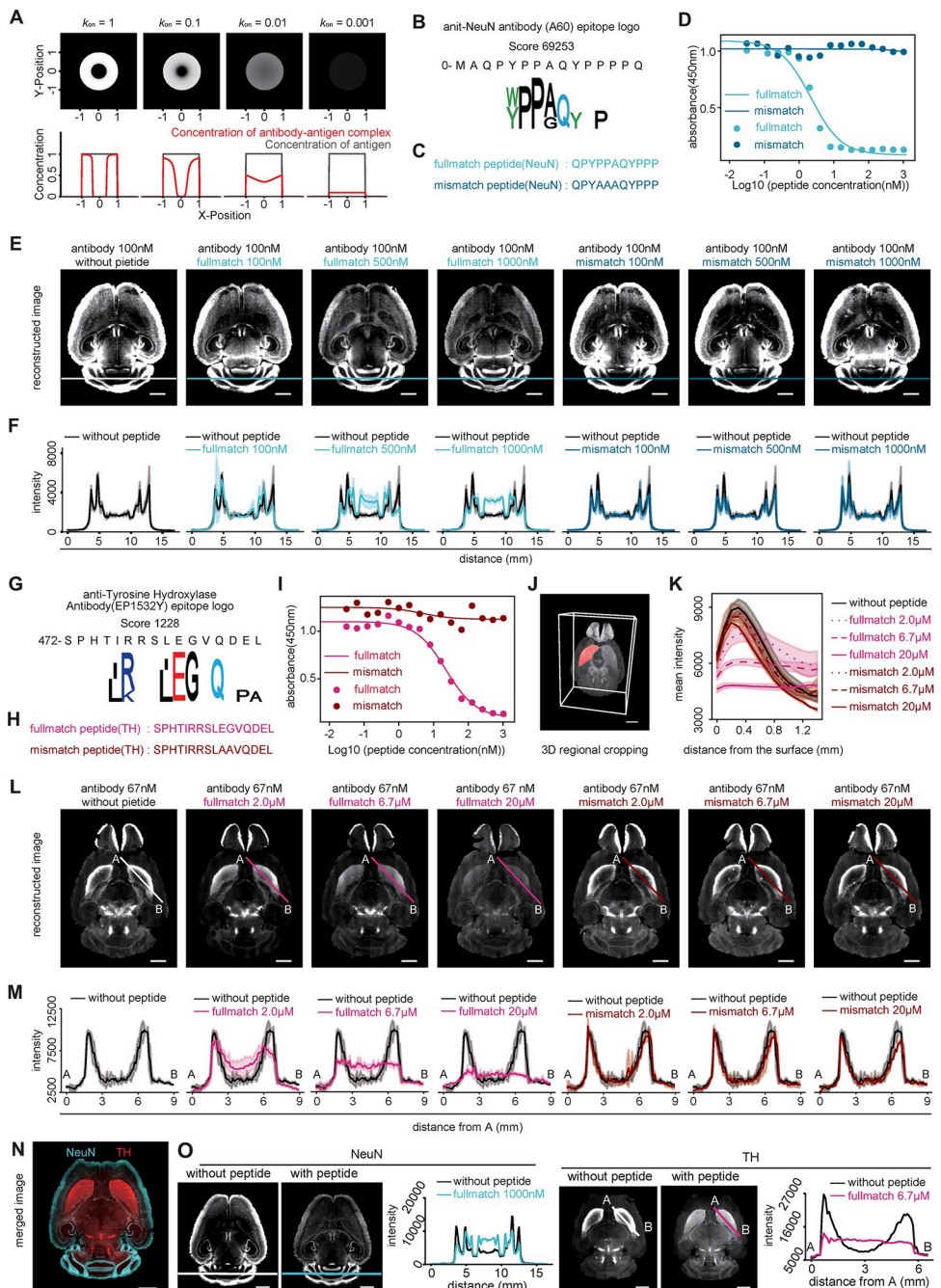

**Fig 5. DECODE improves the penetration speed of antibodies and contributes to uniform staining in 3D immunostaining.** (A) Simulation of the penetration of the antibody for evaluating the 3D staining pattern. The upper panels show staining patterns at the different values of the association rate constant ($k_{on}$). The lower graph shows the concentration of antibody-antigen complex (red) and antigen (gray) in the cross-sectional region at Y = 0. (B, G) Epitope sequence logos of the anit-NeuN monoclonal antibody (clone A60) (B) and anti-tyrosine hydroxylase (TH) antibody (clone EP1532Y) (G). (C, H) Synthesized epitope peptides of A60(c) and EP1532Y(H) based on (B, G). Upper and lower respectively show a wild-type sequence and double mutant. (D, I) Competitive ELISAs for A60 (D) and EP1532Y (I) with full-match or mismatch peptides against target protein. Data are shown as means ± STD ($n = 3$). Lines represent fitting with Michaelis–Menten equation. (E, L) Immunostained images of the sagittal plane of the registered mouse brain. Black, light color, and deep color indicate no peptide, full-match peptide, and mismatch peptide, respectively. Immunostaining was performed by A60 (100 nM) (E) or EP1532Y (67 nM) (L) with various concentrations of epitope peptide. The color lines on the cerebellum in the images of (E) show cross-sectional locations. The color lines from A to B on the striatum of (L) show cross-sectional location. Scale bars, 2 mm. (F, M)

Signal intensities of the cross-section by the cross-sectional location in (E, L). Data are shown as means ± STD ($n = 3$). (J) 3D regional cropping of the striatum. Scale bars, 2 mm. (K) Profiles showing the mean intensity of each depth from the surface of the striatum indicated as the red region in (J). Means ± STD ($n = 3$). (N) Co-staining of NeuN and TH using full-match epitope peptides at optimal concentration. Scale bars, 2 mm. (O) The images show the co-stained sample with or without epitope peptides. Scale bars, 2 mm. Graphs show the mean intensity of the cross-section by the line in the upper images. The data underlying for panels A, B, D, F, G, I, K, M, and O shown in the figure can be found in S1 Data or https://doi.org/10.5281/zenodo.14286317. ELISA, enzyme-linked immunosorbent assay.

beads (Fig 6A) and performed DECODE for these antibodies from the EAE mice ($n = 8$) and healthy mice ($n = 6$). After the second and third rounds of DECODE, some motifs on the immunized peptide were detected only in EAE mice (Fig 6C and 6D). Moreover, these epitopes showed high DECODE scores among all mouse proteins (Figs 6E and S13). In addition, the maximum DECODE scores and ranks of the MOG protein were significantly higher in EAE mice than in healthy mice, as determined by the Mann–Whitney U test (Figs 6F and S12D–S12F). The presence of anti-MOG35-55 antibodies in the serum of EAE and healthy mice was confirmed by ELISAs (Figs 6G and S12G). These results indicate the feasibility of re-identifying pathogen-specific epitopes from blood antibodies using the DECODE method, without relying on the antigen itself.

## Discussion

Epitope analysis using DECODE has simultaneously achieved high-resolution, comprehensiveness, and genome-wide cross-reactivity estimation. This straightforward protocol can potentially enhance throughput using multiwell plates and automatic pipetting machines, and holds promise for comprehensive epitope analysis of commercially available antibodies, which are increasing dramatically every day (Figs 1, 2, S1 and S2). DECODE can quantitatively determine the variations in antigen recognition of antibodies, including information on hot spot residues, and the ELISA results have confirmed its accuracy (Figs 3, S3 and S4). DECODE scores deeply read by NGS may be useful for computational epitope prediction as training data derived from actual antigen-antibody reactions [32,33]. DECODE also predicts antibody cross-reactivity against all proteins across the genome; therefore, it will be useful in avoiding the risk of antibodies binding to non-targets during immunostaining in basic research and pathological diagnosis. In this study, by comprehensively analyzing the epitopes of 230 antibodies, we discovered several identical epitopes among different clones (Figs 4A–4F, S5 and S6). These antibodies can interfere with multicolor staining and sandwich methods in experiments such as immunohistochemistry, western blotting, and ELISA. Without epitope information, these issues cannot be addressed, leading to cost and time wastage in current biological and biomedical research. Similarly, DECODE revealed that different lots of polyclonal antibodies prepared by different individuals had mostly different epitopes (Figs 4G, 4H and S7). These results suggest that using different lots of polyclonal antibodies is equivalent to using different antibodies, and this is the cause of the low reproducibility of experimental data using polyclonal antibodies, which has been pointed out so far.

In recent years, methods for epitope analysis based on peptide selection using 96-well plate throughput, such as AbMap [34] and IMUNE [35], have been reported. Compared to these methods, the DECODE method has a library size that is more than 3 orders of magnitude larger. This large library size may be advantageous in antigen-independent, single-amino acid resolution epitope motif analysis and in pathogenic epitope analysis targeting highly diverse blood antibodies. For example, AbMap addresses the diversity of serum antibodies by enriching and purifying pathogen-specific antibodies using antigen proteins before performing peptide selection. While this improves the accuracy of epitope motifs, it makes it challenging to

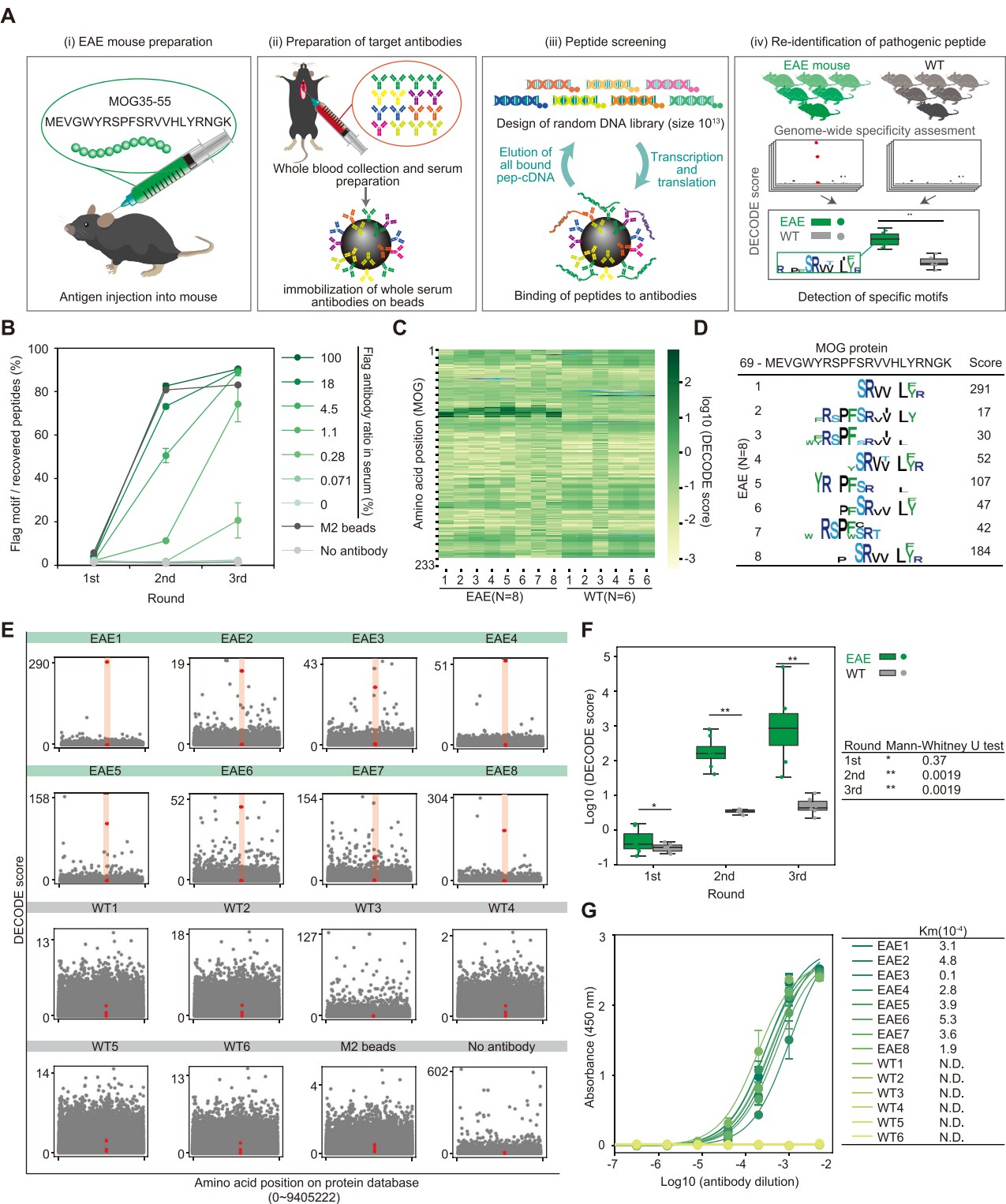

**Fig 6. DECODE identifies the pathogenic epitope for serum antibodies in EAE mice.** (A) Workflow of unbiased epitope analysis of serum antibodies using DECODE. (i) EAE mice preparation by injecting MOG35–55 peptide with CFA. (ii) Immobilization of the serum antibodies on protein G magnetic beads. (iii) DECODE selection. (iv) Comparison of the DECODE results between patient and healthy mouse. (B) Evaluation of the sensitivity of the DECODE was performed using sera added with various concentrations of anti-Flag antibody. The plot shows percentages of Flag motifs in each selection round. Data are shown as means ± STD (*n* = 3). (C) DECODE scores on MOG protein of each EAE mouse serum and healthy mouse serum. (D) Epitope sequence logos at third DECODE selection round. (E) Manhattan plots of the DECODE scores on mouse protein database.

Each panel indicate individual serum antibody (EAE mouse ($n$ = 8), healthy mouse (WT, $n$ = 6), anti-FLAG M2 magnetic beads (M2 beads), and no antibody immobilized beads (No antibody)) in the second DECODE selection round, respectively. MOG protein is shown in red. (F) Box-whisker plot showing the highest DECODE score on MOG protein for each DECODE selection round comparing EAE (green, $n$ = 8) and WT (gray, $n$ = 6). $P$ values were obtained by Mann–Whitney's U test. (G) Binding assays by direct ELISAs for serum antibodies against MOG peptide. Data are shown as means ± STD ($n$ = 3). Lines represent fitting with Michaelis–Menten equation. Table shows calculated Km values. The data underlying for panels B–G shown in the figure can be found in S1 Data or https://doi.org/10.5281/zenodo.14286317. CFA, complete Freund's adjuvant; DECODE, Decoding Epitope Composition by Optimized-mRNA-display, Data analysis, and Expression sequencing; EAE, experimental autoimmune encephalomyelitis; ELISA, enzyme-linked immunosorbent assay; MOG, myelin oligodendrocyte glycoprotein.

explore unknown antigens due to its dependence on antigen proteins. Both IMUNE and DECODE have the potential to explore unknown antigens by directly binding the peptide library to blood antibodies without using antigen proteins. However, for epitope prediction at 1 amino acid resolution, taking into account the importance of hotspot residues and the possibility of amino acid substitutions with similar side chain characteristics, DECODE may be expected to give more accurate results due to its larger library size. The DECODE method has a throughput comparable to AbMap and IMUNE. However, because the protocol proceeds from transcription and translation to peptide extraction with a simple pipetting operation, further high-throughput can be achieved through automation using an automated liquid handler. Additionally, the reaction volume of DECODE is smaller compared to phage/bacterial display, making it suitable for 384-well plate scales, allowing for further parallelization and acceleration. In DECODE analysis, each peptide sequence obtained by NGS is compared one by one with the protein database to determine similarity. Compared to epitope analysis methods that perform alignment and clustering within the collected peptide group, this reduces the risk of missing antigen-derived peptides due to classification bias or discarding peptides with low read numbers. This is particularly effective when targeting polyclonal or blood antibodies. It also has the advantage of predicting genome-wide cross-reactivity. However, there is a possibility of missing sequences not registered in the protein database, which is the limitation of the DECODE method.

Despite the vast and growing number of commercially available antibodies, most epitope information remains unknown. Consequently, immunostaining conditions and antibody selection have been optimized through empirical trial and error. By comprehensively clarifying and disclosing antibody epitope information using DECODE, we can design experimental conditions and select suitable antibodies based on scientific evidence under understanding the recognition patterns, specificity, and cross-reactivity of the antibodies. In fact, in this study, we were able to select the optimal antibody for formalin fixation conditions and antigen retrieval treatment based on the characteristics of the epitope, particularly the amino acids contained in hotspot residues (S8 and S9 Figs). Thus, the detailed epitope information provided by DECODE is expected to enhance the reproducibility and reliability of basic research and medical tests that rely on antibodies.

For the commercial antibodies targeted in this study, each antibody producer provides information about species cross-reactivity on their websites (S1 Table: "Species Reactivity"). For example, anti-c-fos antibodies (2H2 and 9F6) are indicated by their respective manufacturers (Abcam and CST) to be cross-reactive with human, mouse, and rat. The epitope motifs of 2H2 and 9F6 identified by DECODE are conserved across these 3 species. On the other hand, SantaCruz lists cross-species reactivity for its anti-p53 antibodies (Bp53-12, DO-1, and DO-7) with human, mouse, and rat, but the hotspot residue D identified by DECODE is not conserved across these 3 species. In contrast, Abcam lists only human reactivity for the same clones (Bp53-12, DO-1, and DO-7). Specifically, for DO-1, non-reactivity with mouse and rat is explicitly stated, which is consistent with the DECODE results. This highlights the risk of

inaccurate cross-reactivity information from manufacturers. By conducting epitope analysis across various species PDBs using the DECODE method, we can contribute to more reliable predictions of species cross-reactivity.

Since the DECODE method uses linear random peptides containing 12 random amino acids, within this range, it was possible to identify both linear and discontinuous epitopes. However, it is challenging to identify conformational epitopes composed of amino acids that are spaced more than 12 residues apart in the primary structure and get closer in the tertiary structure using the DECODE method. When analyzing conformational epitopes composed of widely separated amino acids using the DECODE method, one approach is to use a structural scaffold with a variable region. Peptide selection using the PURE system can express proteins with lengths of several hundred amino acids, such as scFv [36]. Additionally, it is possible to add chaperones to assist with folding. While it may be possible to use these options for conformational epitope selection, the diversity of unknown conformational epitopes is enormous, and the non-focused selection approach is not expected to be easy. Regarding research antibodies, by examining the manufacturer's website, it is evident that most commercially available antibodies can be used in western blotting. In other words, many commercially available antibodies bind to linear proteins that have been denatured by SDS-PAGE, so a comprehensive identification of the linear epitopes of these antibodies could be highly valuable. At the moment, using the DECODE method to identify epitopes with posttranslational modifications (PTMs), such as phosphorylation and methylation, also presents a challenge. However, if a strategy to incorporate modified amino acids into proteins during translation through genetic code expansion (GCE) is developed, it may be possible to extend epitope analysis to target antibodies that specifically recognize PTMs [37].

Recently, with the advancement of tissue-clearing techniques and light sheet microscopy, 3D immunostaining of large specimens, including whole mouse brains and organs, has garnered increasing attention. However, in 3D immunostaining, most antibodies bind only to the surface of the tissue due to their high affinity, making it difficult for them to penetrate deeper. Several methods for infiltrating antibodies into deep tissues have been reported. However, each method requires the regulation of conditions such as pH, salt concentration, temperature, staining time, and additives [29,38–40]. This makes simultaneous staining with multiple types of antibodies having different optimal conditions challenging. Furthermore, while almost all approaches control antibody-antigen interactions through buffer conditions or temperature, these nonspecific approaches may also impact antibody stability. This concern can limit the development of immunostains for large tissues, such as centimeter-sized tissue blocks or whole primate organs, which require long staining times. In this study, we succeeded in increasing the permeation rate of antibodies by specifically controlling the apparent $k_{on}$ of antibodies using each competitive peptide identified by DECODE (Figs 5, S10 and S11). This strategy allows for simultaneous multicolor staining because the competing peptide affects only its complementary antibody and increases the rate of penetration.

DECODE was able to extract antigen information from a highly diverse mixture of serum antibodies with a sensitivity of less than 1%. This high sensitivity enabled us to successfully re-identify pathogenic peptides in EAE mice (Figs 6, S12 and S13). Conventionally, to respond to the diversity of antibodies in the blood, libraries were constructed based on sequence information such as gene and protein databases of known antigens, or antigen-specific antibodies were purified from the blood and peptides were eluted by using antigenic proteins [34,41–43]. However, such antigen-dependent strategies are not effective for exploring completely unknown antigens. In contrast, DECODE uses a completely random library, directly binds blood antibodies to magnetic beads, and recovers all bound peptides through heat treatment. This strategy, which does not rely on antigenic information, may be effective in identifying unknown

antigens. With this strategy, nonspecific peptides may also be recovered and amplified as noise. We have overcome these drawbacks by performing negative selection with beads that have no immobilized antibodies. In recent years, BCR repertoire analysis has revealed several disease-specific factors. However, this strategy remains challenging due to the intramolecular diversity of antibodies between individuals. Since DECODE can directly obtain antigen information, it has the advantage of being less susceptible to individual differences. Because DECODE can quantitatively and directly elucidate antigen, it has the potential to contribute to identifying the causes of blood-borne infectious diseases, autoimmune diseases, and tumor antigens, and to detailed and quantitative analysis of vaccine efficacy.

## Materials and methods

### Design and preparation of the template DNA library

Single-strand template DNA library CCTAATACGACTCACTATAGGGTTAACTTTAAGAAGGAGATATACATATG(NNK)nTGCGGCAGCGGCAGCGGCAGCTACTTTGATCCGCCGACC ($n$ = 12) was obtained from FASMAC Inc. Further, 25 pmol of the ss-DNA library was formed duplex and amplified using Phusion High-Fidelity DNA Polymerase (NEB), according to the manufacturer's recommended protocol with 0.5 μm Primer 1 (CCTAATACGACTCACTATAGGGTTAACTTTAAGAAGGAGATACATATATG) and 0.5 μm Primer 2 (ggTCGGCGGATCAAAGTAG, g = 2′-O-methylguanosine). The PCR conditions were as follows: 500 μl scale and 4 cycles of temperature changes (95˚C for 10 s, 58˚C for 10 s, and 75˚C for 30 s in a thermal cycler). The PCR products were electrophoresed (135 V, 15 min) on a 2% agarose gel containing 0.1% ethidium bromide to validate the amplification of the DNA library and were purified with Ampure XP. The control DNA template for the anti-Flag antibody (mc1') was prepared based on a previous study [44] (S2 Table).

### Preparation of the mRNA library and puromycin-linked mRNA library

The purified template DNA library was transcribed with the following transcription mixture: 40 mM HEPES-KOH (pH 7.6), 20 mM $MgCl_2$, 2 mM spermidine, 5 mM DTT, 12.5 mM NTPs, and 0.43 μg/μl of the T7 RNA polymerase. The condition of the transcription reaction that can finally inactivate the T7 RNA polymerase was as follows: 37˚C for 1 h, 75˚C for 5 min. All DNA purified above was used in the first round, and 5 μl (approximately 10 pmol) of DNA was utilized in the second and subsequent rounds. The transcription products were annealed with the following mixture: 5 μm Pu-DNA (5′-[PHO] CTCCCGCCCCCCCGTCC [SpC18] 5CC [Puromycin]) and 5 μm splint DNA (5′-GGGCGGGAGGGTCGGCGGATCAA) (S2 Table). The condition of the annealing reaction was as follows: 95˚C for 1 min and 75˚C for 30 s. Then, the temperature was lowered to 25˚C with a constant gradient of 1˚C/15 s. The annealed product was added with 35 U of T4 DNA Ligase (Takara Bio Inc.) and incubated at 37˚C for 1 h and then at 16˚C for O/N. Transcription and ligation products were electrophoresed on 10% acrylamide gel containing 7 M urea (180 V, 40 min) to confirm transcription reaction and ligation efficiency. The Pu-DNA-linked mRNA library of the first round was purified with RNAClean XP. After the second round, this purification step was skipped. The first round of transcription was performed at a 500 μl scale. After the second round, it was performed at a 10 μl scale. The first round of ligation was performed at a 500 μl scale. After the second round, it was performed at a 10 μl scale. Transcription and ligation products derived from the DNA template with and without Gm modification were electrophoresed (180 V, 3 h) on a 30 cm length of 10% acrylamide gel containing 7 M urea to validate transcription reaction and ligation efficiency in a single-base resolution.

### Preparation of the peptide library

The Pu-DNA-ligated mRNA library solution was translated using the PURE system without further purification. In the case of targeting commercial antibodies, peptide translation was performed by the customized PURE system (eliminated RF1 and T7 RNA polymerase). For the serum, peptides were translated by PURE flex1.0 (GeneFrontier Corp.).

### Preparation of the antibody-immobilized magnetic beads

In total, 0.2 to 1 µg of purified antibodies were immobilized on 2.5 µl of Protein G Mag Sepharose (Cytiva) with 5 µl of the following buffer (50 mM Tris-HCl (pH 8.0), 500 mM NaCl, 1% Triton, and 0.01% Tween-20) for 30 min at r.t. The antibody-immobilized beads were washed 10 times with 100 µl of wash buffer (50 mM Tris-HCl (pH 8.0), 500 mM NaCl, 1% Triton X100, and 0.01% Tween-20). Further, 200 µl of the mouse serum was immobilized on 25 µl of Protein G Mag Sepharose (Cytiva) under similar conditions.

### Binding of peptide library to antibody-immobilized magnetic beads

In total, 11.9 µl of translation product, 25 µl of binding buffer (50 mM Tris-HCl (pH 8.0), 10 mM EDTA), and antibody-immobilized magnetic beads were mixed and shaken for 30 min at r.t. Then, the magnetic beads were washed 10 times with a wash buffer. In the mouse serum, this step was performed under the same conditions as mentioned above with 10 times the amount of the reagents.

### Reverse transcription

Purified magnetic beads were added based on the following reverse transcription mixture (0.2 mM dNTPs, 10 mM DTT, and 0.1 µm RT-Primer [P2_ver2] and Proto Script II RTase [New England Biolabs]) and were incubated at 37˚C for 40 min.

### Elution

Magnetic beads were washed with 25 µl of phosphate buffer (50 mM, pH 7.6) and heated at 95˚C for 3 min with 10 µl of phosphate buffer. The supernatants were collected and mixed with the subsequent supernatants washed twice with water as recovered cDNA libraries.

### Avoiding RNA degradation

When implementing mRNA display, it is crucial to prevent mRNA degradation. In this study, we meticulously maintained the experimental equipment with RNase Quiet (Nacalai) to remove nucleases and rinsed RNase-free water. Additionally, we carefully adjusted the pH and quenched the metal ion activity with EDTA as quickly as possible after reaction to avoid RNA degradation.

### Parallelization of DECODE selection

In order to perform epitope screening for 230 antibodies, FastGene 96-well PCR PLate Ultra Easy Cut (0.2 ml/Non Skirted) (Nippon Genetics) was used for the steps of PCR, in vitro transcription, ligation, and translation. Nunclon Sphera 96-Well, Nunclon Sphera-Treated, U-Shaped-Bottom Microplate (Thermo Scientific) was used for the steps of binding to antibody beads, washing, and extraction. All steps were performed manually using a multi-pipette. The steps of binding to antibody beads and washing were performed by adding buffer, shaking

with a shaker, and removing the buffer. One researcher needed 5 days to perform 3 rounds of peptide selection for 96 antibody samples as described above.

## Quantification of the recovered cDNA library

cDNA libraries were quantified via qPCR with TB Green Premix Ex Taq GC (Perfect Real Time) samples containing 1 × Power SYBR Green Master Mix (Applied Biosystems), 0.2 μm primers (forward: CCTAATACGACTCACTATAGGGGTTAACTTTAAGAAGGAGATATA CATATG, reverse: GGTCGGCGGATCAAAGTAG) and 0.5 μl of cDNA library at a volume of 7.5 μl. The PCR conditions were as follows: 1 min at 95˚C, then 40 cycles of 10 s at 95˚C and 30 s at 60˚C.

## Preparation of NGS libraries

In total, 2 μl of the recovered cDNA library was labeled with barcode sequence via PCR using barcode primers (S3 Table). PCR was performed under a similar condition during the amplification step described above. The barcode-labeled cDNAs were mixed and purified with Ampure XP. Sequences were obtained via NGS (Genome Analysis Office, Genetic Information Research Center, Institute for Microbial Diseases, Osaka University) using Illumina MiSeq, HiSeq 3000, or NovaSeq 6000 with approximately 1 to 10 million reads per antibody.

## GPU-based analysis

The sequence similarity was calculated using the optimized BLOSUM62 table, which has negative values corrected to zero as it targets random libraries. The genome-wide protein database of mouse or human, derived from UniProt, was used to calculate the similarity scores and the peptide frequency list, and to derive the inverse cumulative probability distribution of ideally random sequences (P) or sequences recovered by the DECODE method (Q). To quantify the difference between P and Q, the Pearson X2 formula was used due to the greatest distance around the high similarity score area, while the smallest distance was observed in the low similarity area. DECODE score calculations are performed on the GPU and were coded using CUDA C/C++.

## Peptide synthesis

Peptides were synthesized with an automated peptide synthesizer (Syro Wave; Biotage) using Fmoc solid-phase chemistry. Synthesized peptides were treated with a cleavage cocktail (trifluoroacetic acid [88.5% v/v], phenol [4.4% v/v], 1,2-ethanedithiol [2.2% v/v], thioanisole [0.4% v/v], and water [4.4% v/v]) to facilitate cleavage and de-protection at r.t. for 3 h. The molecular weight of the synthesized peptides was confirmed using MALDI-TOF Mass (MALDI-TOFMS AXMA ASSURANCE [Shimadzu]) with α-cyano-4-hydroxycinnamic acid matrix (Sigma-Aldrich). Synthesized peptide sequences for ELISA are shown in S4 Table.

## Confirmation of epitopes by ELISAs

ELISAs of anti-c-Fos monoclonal antibodies (2H2, 9F6, and C-10) and anti-p53 monoclonal antibodies (Bp53-12, DO-1, DO-7, Pab1801, Pab240, and DO-2) to peptides detected using DECODE was performed as follows: 12.5 μl per well of 4 mM peptide solutions were coated on 384-well ELISA plates (Thermo Fisher Scientific) by shaking at r.t. for 1 h. After the plate was stored overnight at 4˚C, it was washed 3 times with PBST (1× PBS with 0.1% Tween 20) and incubated in 120 μl per well of blocking buffer (Blocking One [Nacalai Tesque] diluted in PBS at a ratio of 1:2) at r.t. for 1 h. After washing 3 times with PBST, 12.5 μl per well of 0.2 to

1 µg/ml monoclonal antibodies in the blocking buffer were added and incubated by shaking at r.t. for 1 h. The plate was washed 3 times with PBST and then incubated with a secondary antibody conjugated with horseradish peroxidase (HRP) (BioRad) diluted with a blocking buffer at a ratio of 1:2,000 by shaking at r.t. for 1 h. After washing 12 times with PBST, 25 µl per well of the HRP substrate (ELISA POD Substrate TMB Kit (Popular), Nacalai Tesque) was added and incubated by shaking at r.t. for 10 to 30 min. The peroxidase activity was discontinued by adding 50 µl per well of 0.1 M $H_2SO_4$. Color development was evaluated at 450 nm on a microplate reader (Powerwave HT Spectrophotometer Microplate Reader, BioTek). c-Fos protein and mutants were used as an HEK293T cell lysate overexpressed with c-Fos protein. In total, 12.5 µl per well of c-Fos protein and mutant solution were coated on 384 plates. The p53 protein was a recombinant human p53 protein (0.65 µg/ml) (Abcam). The NeuN protein was from the brain cell lysate.

## Assessment of antigen retrieval using ELISAs

In total, 12.5 µl per well of c-Fos protein was coated on two 384 plates. Next, 12.5 µl of 100 mM glycine in 4% paraformaldehyde/PBS was added and stayed at 35˚C for 48 h. One of the plates was treated with heat-induced epitope retrieval in 2× SSC Buffer (GeneMark) at 98˚C for 45 min. However, the other plates were not. In total, 12.5 µl per well of 0.2 to 0.3 µg/ml anti-c-Fos antibodies (2H2 or 9F6) in the blocking buffer was incubated. Washing, blocking, incubation, and TMB-HRP chromogenic reaction was performed similarly to the abovementioned session.

## Mass spectrometry

In-solution protein digestion was performed based on the phase-transfer surfactant protocol [45]. Antibody (Ep53-12 or DO-1, 100 ng each), cytochrome c from the equine heart (40 ng), and lysozyme from egg white (40 ng) were solubilized in 20 µl of phase-transfer surfactant lysis buffer (50-mM NH4HCO3, 12 mM sodium deoxycholate, and 12 mM sodium N-lauroylsarcosinate) and were incubated at 95˚C for 5 min. Cysteine reduction and alkylation were performed via incubation with 20 mM tris ((2-carboxyethyl) phosphine) and 20 mM iodoacetamide at 37˚C for 1 h in the dark. Then, quenching was conducted with 20 mM dithiothreitol at r.t. for 20 min. The sample was digested with 40 ng of trypsin (proteomics grade, PROMEGA) at 37˚C overnight. Via the addition of 1% TFA and centrifugation at 10,000 g for 10 min, the reaction was stopped, and the detergents were removed. In addition, in-gel protein digestion was performed using the gel bands of heavy and light chains of 800-ng antibodies (Bp53-12, C-11, D11, DO-1, DO-2, DO-7, Pab1801, and Pab240) separated via sodium dodecyl-sulfate polyacrylamide gel electrophoresis. The tryptic digest was loaded on a C18 pipette tip prepared, as reported in a previous study [46]. Then, dimethyl labeling was performed according to a previous report [47]. Formaldehyde ($CH_2O$, Sigma-Aldrich) or isotope-labeled formaldehyde (13CD2O, ISOTEC) was used to light or heavy label the sample with each antibody. The dimethyl-labeled peptides on the tip were eluted with 80% acetonitrile and 0.1% TFA and dried using SpeedVac. Then, they were stored at −80˚C until mass spectrometry. The dimethyl-labeled peptides were reconstituted by 2% acetonitrile and 0.1% TFA. Then, equal amounts of light- and heavy-labeled samples were mixed just before MS measurement. The mixture was analyzed with the Orbitrap LTQ Velos mass spectrometer (Thermo Scientific) and the UltiMate 3000 RSLCnano-flow HPLC system (Thermo Scientific) under CID MS/MS mode. Raw data were searched with MaxQuant (version 2.0.1.0) against the UniProt mouse database (17,082 genes downloaded from Swiss-Prot in 2021.7) combined with the mouse immunoglobulin genes (1,213 genes from TrEMBL and 861 genes from VBASE2) [48].

Peptide identification was filtered at FDR 1%, and carbamidomethylation of Cys was set under a fixed modification. Oxidation of Met and determination of Asn/Gln was set for variable modifications. The MS intensities of the peptides were obtained via label-free quantification implemented in MaxQuant [49].

## Mice

All experimental procedures and housing conditions related to the mice were approved by the Institutional Animal Care and Use Committee of the RIKEN Center for Biosystems Dynamics Research and the University of Tokyo (QA2013-01-31). All animals were cared for and treated humanely, in accordance with the Institutional Guidelines for Experiments using Animals. The mice had ad libitum access to food and water and were kept under ambient temperature and humidity conditions, following a 12-h light-dark cycle. All C57BL/6N and B6N mice were procured from CLEA Japan, located in Tokyo, Japan. The experiments were conducted at the RIKEN Center for Biosystems Dynamics Research or the University of Tokyo.

## Three-dimensional whole-brain staining using the CUBIC protocol

For whole-brain staining, the brains of adult mice (C57BL/6N) were used. MK-801 was injected into the mice. Dissection, fixation, PBS washing, tissue clearing, staining, imaging, and analysis were performed according to previous studies [29,30]. For immunostaining in Fig 6, the respective concentrations of peptide were used instead of the quadrol. For each experiment, tissue clearing imaging was confirmed to be reproducible on individual mouse whole brain samples ($n = 3$).

## Preparation of the serum of mice with EAE

Via the subcutaneous injection of MOG peptide (UT) antigen at 2 back sites with CFA, 10- to 12-week-old B6N mice ($n = 8$) were generated. Pertussis toxin was administered on days 0, 2, and 7. After achieving a score of $\geq 3$, the whole-blood sample was collected to obtain the serum [50].

## Supporting information

**S1 Fig. Detailed improvements for high-throughput and detailed epitope analysis, related to Fig 1.** (A) Template DNA design of DECODE selection. Yellow, blue, green, pink, and gray indicate a T7 promoter, a Shine–Dalgarno sequence, a start codon, a random region, and a linker region, respectively. N and K indicate a mixed nucleotide of A, T, G, C and T, G. NNK is repeated 12 times. (B) Detailed analysis of ligation between Pu-DNA and transcribed mRNA without Gm modified DNA template. Samples are separated in single-nucleotide resolution with a large urea PAGE (10% AA and 6M Urea, 30 cm in length). (C) Schematic illustration of mRNA preparation using 2′-O-methylguanosine (Gm) modified templated DNA. Red spheres indicate Gm. The antisense strand of the 5′ end of the template DNA was modified by 2 Gms to reduce the run-off activity of T7 RNA polymerase. (D) Two kinds of commonly used ligation forms between RNA and Pu-DNA in mRNA display methods are illustrated. The circle included "P" is Puromycin. Linear form ligation is suppressed to a low efficiency by run-off products. (E) Ligation efficiencies of hairpin and linear form using urea PAGE (10% AA and 6 M Urea). (F) Translation efficiencies of hairpin and linear forms using northern blotting (e-PAGEL gradient gel (10% to 20%), nylon membrane (Hybond N+)) and calculated by ImageJ. (G) Quantification of the recovered peptide-cDNA complex using anti-Flag antibody magnetic beads (M2 beads). The recovered complexes used were quantified by qPCR as a threshold

cycle (Ct). Data are shown as means ± STD ($n$ = 2). The data underlying for panels G shown in the figure can be found in S2 Data or https://doi.org/10.5281/zenodo.14286317.
(PDF)

**S2 Fig. Method for A GPU-based calculating the probability of amino acid occurrence in a genome-wide protein database, related to Fig 2.** (A) One of the probability distributions on each similarity score for a completely randomized DNA library. This plot shows the binding site (22–33 aa) of anti-c-fos antibody clone 9F6. (B) Distributions of the experimental probabilities of the similarity scores about clone 9F6 on the same position with A. (C) Plots of the distance at each similarity score between P and Q of the binding sites are calculated by various distance functions. (D–H) DECODE score plots on the mouse c-fos protein for monoclonal anti-c-fos antibodies and without antibodies. (I) Manhattan plot of anti-c-fos antibody clone 9F6 against all mouse proteins, which is calculated with a modified WAC table. Red dots and highlights indicate the c-fos protein. This plot was visualized with downsampled data to 1/50 using LTTB. (J) Top 100 DECODE scores of (I). Red dots indicate the c-fos protein. (K) Comparison of the DECODE score distribution on the mouse c-fos protein for anti-c-fos antibody clone 9F6 between modified BLOSUM62 and modified WAC. (L, M) Amino acid similarity tables used for DECODE analysis. Modified BLOSUM62 (L) and modified WAC (M). The data underlying for panels A–M shown in the figure can be found in S2 Data or https://doi.org/10.5281/zenodo.14286317.
(PDF)

**S3 Fig. Reliability of binding sites identified by DECODE, related to Fig 3.** (A) Epitope logos at the highest DECODE score position on the c-fos protein at the third round of DECODE selection for anit-c-fos monoclonal antibodies (9F6, 2H2, and C-10) in independent experiments ($n$ = 3). (B) Reproducibility of the DECODE selection of 3 kinds of monoclonal anti-c-fos antibodies (clone 2H2, 9F6, and C-10) during independent experimental ($n$ = 3). (C) Pairwise correlation of the DECODE scores between independent experiments ($n$ = 3) or different clones. (D) The upper table shows synthesized peptides. Lower bar graph showing direct ELISA signals of A450 for each anti-c-fos antibody against upper peptides. Data are shown as actual data and means ± STD ($n$ = 3). (E) Wild-type and double mutant c-fos protein sequences. (F, G) Direct ELISA results against wild-type and each mutant c-fos proteins by 3 anti-c-fos antibodies. Data are shown as means ± STD ($n$ = 3). (H) Binding curves of (G). Lines represent fitting with Michaelis–Menten equation. The table shows the saturation values of each curve. (I) Direct ELISA for binding anti-c-fos antibody (C-10) to antigens derived from the 4 proteins (AHNAK2, LAMA1, FOSL2, and PLCD4) predicted to cross-react in Fig 3E. The left table lists each antigen and the Km values calculated from the binding curves. The peptide antigens were chemically synthesized based on the sequences of the regions on each protein with the highest DECODE scores. The binding curves (center) are for the peptide antigens, and the binding curves (right) are for the protein antigens. The binding curves were fitted using the Michaelis–Menten equation. Data are shown as mean ± STD ($n$ = 3). The data underlying for panels A–I shown in the figure can be found in S2 Data or https://doi.org/10.5281/zenodo.14286317.
(PDF)

**S4 Fig. Accuracy of binding hotspot residues identified by DECODE, related to Fig 3.** (A) The bar graphs show raw data of Fig 3C. (B) The bar graph shows the logarithm of the IC50 of each mutant peptide calculated from the data in Fig 3C using the Michaelis–Menten equation. (C) Epitope sequence logo (upper) and Manhattan plot on the human protein database for anti-NeuN antibody (clone A60) (Lower). Red circles indicate NeuN protein. (D) Competitive

curves of anti-NeuN antibody (clone A60) against NeuN protein inhibited by single amino acid mutant peptides at each concentration. A450s were normalized without competitive peptide conditions. Data are shown as means ± STD ($n$ = 3). Lines represent fitting with Michaelis–Menten equation. Mutant peptide sequences are provided as S2 Table. (E) Scatter plot of the correlation about anti-NueN antibody (clone A60) between the difference in converged amino acid ratios between wild and mutant amino acids (Δconvergence ratio (WT-MT)) and the difference of IC50 calculated in S4D between mutant and wild type ($Log_{10} \Delta IC_{50}$ log (WT-MT)). Plots, lines, and shaded areas represent actual data, the regression line and the 95% confidence bounds, respectively. Spearman's correlation coefficient was R = 0.79. (F, G) Results of the competitive ELISAs and the logarithmic IC50s of each single amino acid mutant peptide for NeuN protein for anti-NueN antibody (clone A60). Data are shown as means ± STD ($n$ = 3). The data underlying for panels A–D and F–G shown in the figure can be found in S2 Data or https://doi.org/10.5281/zenodo.14286317.
(PDF)

**S5 Fig. Discovery of epitope similarities among distinct clones, related to Fig 4.** (A) Sequence logos of the most converged peptide in round 3 of DECODE selection for anti-p53 monoclonal antibodies (clone Bp53-12, DO-1, and DO-7) in independent experiments ($n$ = 3). (B) Manhattan plots of each anti-p53 antibody on the human protein database. Plots of the p53 protein are shown in red. (C) Competitive ELISAs of anti-p53 antibodies (clones Bp53-12, DO-1, and DO-7) with single amino acid mutant peptides at each concentration against p53 protein. A450 was normalized by the values without a competitive peptide. Plots indicate means ± SD of independent experiments ($n$ = 3). Lines represent fitting to Michaelis–Menten equation. Mutant epitope peptides are provided as a peptides list (S2 Table). (D) Bar graphs show raw data of (C). (E) The bar graph shows the logarithm of the IC50 of each mutant peptide calculated from the data in (C) using the Michaelis–Menten equation. The data underlying for panels A–E shown in the figure can be found in S2 Data or https://doi.org/10.5281/zenodo.14286317.
(PDF)

**S6 Fig. Confirmation of differences in antibody sequences of monoclonal antibodies with similar epitopes, related to Fig 4.** (A) Direct ELISA results for anti-p53 antibodies (clones Bp53-12, DO-1, DO-7, Pab1801, Pab240, and DO-2) against each peptide shown in Fig 4C, c-fos protein and BSA. Data are shown as means ± STD ($n$ = 3). (B) ELISA titration results in anti-p53 antibodies (clones Bp53-12, DO-1, and DO-7) against p53 protein. Data are shown as means ± STD ($n$ = 3). (C) Scatter plots of the MS intensities of tryptic peptides derived from Bp53-12 versus 4 p53 antibodies (clones Bp53-12, DO-1, DO-7, and Pab240). Orange, gray, yellow, red, and blue circles indicate IgG, IgG2A, IgG2B, Ig kappa, and unidentified, respectively. (D) Scatter plots of the MS intensities of tryptic peptides derived from Bp53-12 versus 4 p53 antibodies (clones Bp53-12, DO-1, DO-7, and Pab240). Orange and blue circles indicate the variable region and the constant region, respectively. The data underlying for panels A–D shown in the figure can be found in S2 Data or https://doi.org/10.5281/zenodo.14286317.
(PDF)

**S7 Fig. Epitope differences for polyclonal antibodies among different lots despite of the same catalog number, related to Fig 4.** (A) Verification of the anti-c-fos polyclonal antibodies at the epitope level using DECODE. Different lot products made by different individuals showed different epitope profiles. (B) Pairwise correlation of the DECODE scores between independent experiments ($n$ = 3) and different lot products. (C) Manhattan plots of the DECODE scores on the human protein database. Each panel indicates about different lots

(cat#ab190289; lot.GR323255-1, GR3313102-1, cat#ab209794; lots.GR3198011-8, GR3266315-7) at third DECODE selection round. Red circles indicate c-fos protein. (D) ELISA results of anti-c-fos polyclonal antibodies against each epitope peptide. Data are shown as means ± STD ($n = 3$). The data underlying for panels A–D shown in the figure can be found in S2 Data or https://doi.org/10.5281/zenodo.14286317.
(PDF)

**S8 Fig. A rationale for optimizing and troubleshooting immunostaining conditions provided by DECODE, related to Fig 5.** (A) Sequence logos of the most converged peptide in round 3 of the DECODE analysis for anti-c-fos monoclonal antibodies (clone 2H2 and 9F6). Amino acids indicated red form irreversible methylene bridge with amine group by formalin or PFA fixation. (B) Verification by ELISA of antigenicity changes of each anti-c-fos antibody depending on the fixation strength and the presence or absence of antigen retrieval. Data are shown as means ± STD ($n = 6$). (C) Workflow of whole-brain immunostaining using anti-c-fos monoclonal antibodies (clone 2H2 or 9F6) and whole-brain analysis of c-fos positive cells. (i) Injection of the drug (MK-801), perfusion fixation, and dissection of the brain. (ii) Antigen retrieval within saline sodium citrate buffer (SSC) at 80˚C for 5 h. Strongly fixed samples were mimicked by PFA + 100 mM Gly. (iii) Tissue clearing and staining with CUBIC-HV method. (iv) 3D imaging with a high-resolution light-sheet fluorescence microscope. (v) Cell detection counting of c-fos positive cells across the whole brain. (D) Representative single-plane brain images of the c-fos signal without (upper) or with (lower) antigen retrieval. Scale bars, 100 μm. (E) Magnified images of the hippocampus. Scale bars, 2 mm. (F) Cell number of c-fos positive cells. Data are shown as means ± STD ($n = 3$). The data underlying for panels A, B, and F shown in the figure can be found in S2 Data or https://doi.org/10.5281/zenodo.14286317.
(PDF)

**S9 Fig. Differences in antigen retrieval of whole mouse 3D immunostaining with different antibodies, related to S8 Fig.** (A) Magnified images of the cortex. Scale bars, 2 mm. (B) Detection of the c-fos positive cells. The red dot indicated detected cells. Scale bars, 2 mm. (C) Other depth images of a single plate in S8 Fig. Scale bars, 100 μm.
(PDF)

**S10 Fig. Antibody permeability simulation and image analysis, related to Fig 5.** (A–D) Simulation of antibody penetration speed changes due to changes in each parameter. The upper panels show staining patterns. The lower line graph shows the concentration of antibody-antigen complex (red) and total antigen (gray) in the cross-sectional region at Y = 0. (E) Schematic of the transformation of whole-brain images for comparative analysis. The nonlinear transformation is calculated using the before-registration structure image (upper left) and the reference structure image using auto fluorescence at the GFP channel (upper right). The nonlinear transformation is applied to the before-registration immunostaining image (lower left) to output the after-registration immunostaining image (lower right). Scale bars, 2 mm. (F) The upper panel shows the merged image before registration (red, before-registration structure image; green, reference structure image). The lower panel shows the merged image after registration (red, after-registration structure image; green, reference structure image). Scale bars, 2 mm. The data underlying for panels A–D shown in the figure can be found in S2 Data or https://doi.org/10.5281/zenodo.14286317.
(PDF)

**S11 Fig. Antibody permeability for 3D immunostaining of mouse whole brain from various aspects, related to Fig 5.** (A) Immunostained images stained by anti-NeuN antibody (clone A60) with or without epitope peptide of the coronal plane of the registered brain. The

color lines in the images show cross-sectional locations. Scale bars, 2 mm. (B) Mean intensities of the cross-section by the line in (A). (C, D) Immunostained images of the other position of S10A Fig. (E) Immunostained images stained by anti-TH antibody (clone EP1532Y) with or without epitope peptide of the coronal plane of the registered brain. The color lines in the images show cross-sectional locations. Scale bars, 2 mm. (F) Mean intensities of the cross-section by the line in S10E Fig. Data are shown as means ± STD ($n = 3$). The data underlying for panels B, D, and F shown in the figure can be found in S2 Data or https://doi.org/10.5281/zenodo.14286317.
(PDF)

**S12 Fig. Re-identification of pathogenic epitopes from serum antibodies in EAE mice, related to Fig 6.** (A) Western blot of mouse serum antibodies and anti-Flag M2 antibody detected by anti-mouse IgG-HRP antibody. (B) The standard curve shows the relative intensity of mouse IgG derived from dilution series of anti-Flag M2 antibodies as shown in (A). The equation was obtained by linear approximation (Intercept = 0, R2 = 0.99). The IgG concentration in 1/10 diluted wild-type mouse serum was determined to be 0.094 μg/μl. (C) Bar plots of Fig 6B. (D) The highest DECODE score (log10) in the MOG protein sequence in each DECODE selection round. (E) The highest rank of the epitope on the MOG protein in the mouse protein database in each DECODE selection round. (F) Box-whisker plot showing the maximum rank on the MOG protein in the mouse protein database comparing EAE (green, $n = 8$) and healthy (gray, $n = 6$). $P$ values were obtained by Mann–Whitney's U test (* = 0.25, ** = 0.0019). (G) Raw data of Fig 6G is shown. The data underlying for panels B–G shown in the figure can be found in S2 Data or https://doi.org/10.5281/zenodo.14286317.
(PDF)

**S13 Fig. Transition of epitope appearance in each round of peptide selection, related to Fig 6.** (A) Manhattan plot of the DECODE scores on the mouse protein database. Red circles indicate MOG protein. The data underlying for panels A shown in the figure can be found in S2 Data or https://doi.org/10.5281/zenodo.14286317.
(PDF)

**S1 Table. Results of comprehensive epitope analysis by DECODE.**
(PDF)

**S2 Table. List of nucleic acid sequences used for peptide selection.**
(PDF)

**S3 Table. List of barcode primer sequences used for NGS.**
(PDF)

**S4 Table. List of synthetic peptide sequences used in ELISA.**
(PDF)

**S1 Data. All numerical values underlying the Figs 1–6.** (XLSX uploaded to Zenodo: https://doi.org/10.5281/zenodo.14286317.)
(ZIP)

**S2 Data. All numerical values underlying the S1–S13 Figs.** (XLSX uploaded to Zenodo: https://doi.org/10.5281/zenodo.14286317.)
(ZIP)

**S1 Raw Images. Raw images of Figs 1B, S1B, S1E, S1F and S12A.**
(TIF)

## Acknowledgments

We thank the people at RIKEN and The University of Tokyo, in particular R.G. Yamada, S. Shi, S. Horiguchi, and T. Asanuma for helping with data analysis; J. Adachi and K. Masuda for helping with PURE system preparation; E.A. Susaki, H. Fujishima, and C. Shimizu for helping with 3D immunostaining; K. L. Ode for helping with the simulation of the antibody penetration.

## Author Contributions

**Conceptualization:** Katsuhiko Matsumoto, Shoko Y. Harada, Hiroki R. Ueda.

**Data curation:** Katsuhiko Matsumoto, Shoko Y. Harada, Shota Y. Yoshida, Ryohei Narumi, Tomoki T. Mitani, Saori Yada, Aya Sato.

**Formal analysis:** Katsuhiko Matsumoto, Shoko Y. Harada, Shota Y. Yoshida, Ryohei Narumi, Tomoki T. Mitani, Aya Sato.

**Funding acquisition:** Katsuhiko Matsumoto, Shoko Y. Harada, Shota Y. Yoshida, Tomoki T. Mitani, Hiroki R. Ueda.

**Investigation:** Katsuhiko Matsumoto, Shoko Y. Harada, Shota Y. Yoshida, Ryohei Narumi, Tomoki T. Mitani, Saori Yada, Aya Sato.

**Methodology:** Katsuhiko Matsumoto, Shoko Y. Harada, Shota Y. Yoshida, Ryohei Narumi, Tomoki T. Mitani, Yoshihiro Shimizu.

**Project administration:** Katsuhiko Matsumoto, Shoko Y. Harada, Hiroki R. Ueda.

**Resources:** Katsuhiko Matsumoto, Shoko Y. Harada, Saori Yada, Aya Sato, Yoshihiro Shimizu.

**Supervision:** Katsuhiko Matsumoto, Eiichi Morii, Yoshihiro Shimizu, Hiroki R. Ueda.

**Validation:** Katsuhiko Matsumoto, Shoko Y. Harada, Shota Y. Yoshida, Ryohei Narumi, Saori Yada, Aya Sato.

**Visualization:** Katsuhiko Matsumoto, Shoko Y. Harada, Shota Y. Yoshida, Ryohei Narumi.

**Writing – original draft:** Katsuhiko Matsumoto, Shoko Y. Harada, Shota Y. Yoshida, Ryohei Narumi.

**Writing – review & editing:** Katsuhiko Matsumoto, Shoko Y. Harada, Ryohei Narumi, Tomoki T. Mitani, Yoshihiro Shimizu, Hiroki R. Ueda.

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
