## [Editor Report · Decision Letter 0]

10 Jun 2024

Dear Dr Ueda, 

Thank you for submitting your manuscript entitled "Genome-wide epitope identification with single-amino-acid resolution via high-throughput and unbiased peptide analysis" for consideration as a Research Article by PLOS Biology.

Your manuscript has now been evaluated by the PLOS Biology editorial staff as well as by an academic editor with relevant expertise and I am writing to let you know that we would like to send your submission out for external peer review.

Once your full submission is complete, your paper will undergo a series of checks in preparation for peer review. After your manuscript has passed the checks it will be sent out for review. To provide the metadata for your submission, please Login to Editorial Manager (https://www.editorialmanager.com/pbiology) within two working days, i.e. by Jun 12 2024 11:59PM.

Kind regards,

Suzanne

Suzanne De Bruijn, PhD, 

Associate Editor

PLOS Biology

sbruijn@plos.org

---

## [Decision Letter · Decision Letter 1]

1 Aug 2024

Dear Dr Ueda,

Thank you for your patience while your manuscript "Genome-wide epitope identification with single-amino-acid resolution via high-throughput and unbiased peptide analysis" was peer-reviewed at PLOS Biology. Please accept my apologies for the delays that you have experienced during the peer review process. Your manuscript has now been evaluated by the PLOS Biology editors, an Academic Editor with relevant expertise, and by two independent reviewers. 

In light of the reviews, which you will find at the end of this email, we would like to invite you to revise the work to thoroughly address the reviewers' reports.

As you will see, the reviewers are generally positive about your epitope mapping method and think it will be useful for the field. However, Reviewer #1 requests that benchmarking assays that provide a side-by-side comparison with existing methods (such as AbMap) are included in the revision, as well as discussing the limitations of the approach. In addition, the reviewers ask that important reporting and methodological details are included in the manuscript. 

After discussions within the editorial team, we think that your manuscript would be a better fit a 'Methods and Resources' Article at the journal. Upon resubmission, we would be grateful if you could please tick 'Methods and Resources' as the article type in the drop down menu in the online submission form. 

Given the extent of revision needed, we cannot make a decision about publication until we have seen the revised manuscript and your response to the reviewers' comments. Your revised manuscript is likely to be sent for further evaluation by all or a subset of the reviewers.

**IMPORTANT - SUBMITTING YOUR REVISION**

*Re-submission Checklist*

*Published Peer Review*

*PLOS Data Policy*

*Blot and Gel Data Policy*

Sincerely,

Richard

Richard Hodge, PhD

rhodge@plos.org

REVIEWS:

Reviewer #1: The manuscript presents DECODE for epitope mapping. This method allows for the identification of epitope patterns recognized by monoclonal and polyclonal antibodies at single amino acid resolution, predicting cross-reactivity against a comprehensive protein database. The study's strength lies in its precision, which addresses the critical need for reproducibility and reliability in antibody-based experiments. The authors successfully demonstrate the application of DECODE by identifying an epitope that corresponds to a peptide inducing an autoimmune disease model in mice without prior knowledge of the antigen. This highlights DECODE's potential in improving diagnostics by providing high-quality epitope information. Interestingly, the integration of DECODE into a novel 3D immunostaining method enhances antibody penetration into tissues, further validating its practical utility. The method's applicability to complex blood antibodies, as shown through serum analysis from mice with experimental autoimmune encephalomyelitis (EAE), showcases its versatility and robustness. In summary, this study offers an advancement in epitope mapping, providing a reliable, high-resolution method that can be widely applied. 

Major concerns:

1. The review of the related studies/ technologies is not sufficient. Highly related methods, such as the phage display based strategy from Epitopic and the AbMap strategy have not been well discussed. A clear comparison among DECODE and these strategies is highly recommended, especially, a sdie-by-side comparison between DECODE and AbMap;

2. It is not clear how many rounds of selection is recommend? If several rounds of selection is required, the samples that could be analyzed in one test should be very limited;

3. The limitations of the DECODE strategy should be sufficiently discussed, such as additional consideration for preparing RNA.

Minor concerns:

1. Line 47, "to demonstrated" should be "to demonstrate";

2. It is known that ~80% of the antibodies are recognizing conformational epitopes, especially the therapeutic antibodies. How to resolve the conformational epitopes? More discussion is needed;

3. Is GPU really necessary for the data analysis?

4. Please clearly define what is the "identified epitope peptides";

5. Please add the scores directly to the logos.

Reviewer #2: In this paper, the authors established a protocol for the random-peptide library selection against antibodies using mRNA display, and reported large-scale epitope mapping for more than 200 antibodies. In addition, (although not directly related to the establishment of this method) they showed that co-addition of epitope peptides enables high-affinity antibodies to penetrate deep into tissues for 3D immunostaining.

Following points should be addressed before publication.

p.7, lines 135-138: The authors state the shortcomings of "conventional mRNA display methods", but because protocols and formats for mRNA display vary among research groups, they should cite the literature for the conventional methods they describe here.

p.8, lines 167-172: The authors show that using the improved mRNA display, the percentage of peptides that bind to the FLAG antibody exceeded 90% after three rounds, but they should show how much the enrichment efficiency improved compared to when the unmodified protocol was used.

p.8, lines 177-179: The authors stated that "Ultimately, it will be possible to perform peptide selection targeting at least approximately 100 individual antibodies per week using multiwell plates, increasing the throughput by more than 100 times using existing methods." When they actually performed the selection against the 230 antibodies shown in Table S1, did they not use 96-well microplates or an automatic pipetting machine? (This is not described in the Method section.) If the authors did it manually, how long did it take? (e.g., more than 230 weeks?)

p.11, lines 248-254: What was the protein that C-10 was predicted to cross-react with in Fig. 3E, right (the DECODE score, 10982)? Have the authors confirmed by ELISA that C-10 actually binds to that protein? Is there any reason to believe that C-10 binding to that protein would affect the results of the immunostaining in Ref. 29?

p.12: The results in Table S1 are very interesting. In particular, I have the following questions:

- For antibodies whose "species reactivity" includes not only human but also mouse and/or rat, are the motifs of the DECODE sequence in the antigen protein conserved in these species? Conversely, for human-specific antibodies, are there mutations in the DECODE site in mouse and rat?

- Are there any characteristics specific to human epitopes that deviate from randomness in the DECODE peptide sequences in Table S1?

- Were three rounds of selection performed for all antibodies in Table S1?

- Are there any commonalities among the 13 antibodies whose DECODE was "N.D"?

Minor points

p4, line 64: The K in "Kd" should be in italics.

p8, line 184: It would be nice if there was a brief explanation of BLOSUM62 (using the example in Fig. 2A bottom).

p14, lines 320, 325, 332: The k in "kon" should be in italics.

p25, line 596: The numbers in "H2SO4" should be subscripted.

p26, line 629: The number in "CH2O" should be subscripted.

pp.29-32: The journal names of references 6, 19, 23 and 33 should be abbreviated.

---

## [Editor Report · Decision Letter 2]

15 Nov 2024

Dear Dr Ueda,

Thank you for your patience while we considered your revised manuscript "Genome-wide epitope identification with single-amino-acid resolution via high-throughput and unbiased peptide analysis" for publication as a Methods and Resources Article at PLOS Biology. This revised version of your manuscript has been evaluated by the PLOS Biology editors and the Academic Editor.

Based on our Academic Editor's assessment of your revision, I am pleased to day that we are likely to accept this manuscript for publication, provided you satisfactorily address the following data and other policy-related requests that I have provided below (A-G). Please note that I have also taken the liberty of changing the article type to 'Methods and Resources' and I am sorry that you encountered issues with this during resubmission:

(A) We routinely suggest changes to titles to ensure maximum accessibility for a broad, non-specialist readership. In this case, we would suggest a minor edit to the title, as follows. Please ensure you change both the manuscript file and the online submission system, as they need to match for final acceptance:

“DECODE enables high-throughput mapping of antibody epitopes at single amino acid resolution”

(B) In the animal ethics statement in the Methods section, please provide the specific approval number that you received from the RIKEN IACUC to conduct the study. 

(C) You may be aware of the PLOS Data Policy, which requires that all data be made available without restriction: http://journals.plos.org/plosbiology/s/data-availability. For more information, please also see this editorial: http://dx.doi.org/10.1371/journal.pbio.1001797

-Supplementary files (e.g., excel). Please ensure that all data files are uploaded as 'Supporting Information' and are invariably referred to (in the manuscript, figure legends, and the Description field when uploading your files) using the following format verbatim: S1 Data, S2 Data, etc. Multiple panels of a single or even several figures can be included as multiple sheets in one excel file that is saved using exactly the following convention: S1_Data.xlsx (using an underscore).

-Deposition in a publicly available repository. Please also provide the accession code or a reviewer link so that we may view your data before publication. 

Figure 1C-D, 2A-H, 3A-E, 4A-H, 5A-B, 5D, 5F, 5I, 5K, 5M, 5O, 6B-G, S1G, S2A-M, S3A-I, S4A-G, S5A-E, S6A-D, S7A-D, S8A-B, S8F, S10A-D, S11B, S11D, S11F, S12B-G, S13A

(D) Please also ensure that each of the relevant figure legends in your manuscript include information on *WHERE THE UNDERLYING DATA CAN BE FOUND*, and ensure your supplemental data file/s has a legend.

(E) Thank you for providing the original, uncropped and minimally adjusted images for the following Figures: Figure 1B, S1B, S1E-F, S12A. However, we ask that you please combine the raw images into a single S1_raw_images file in the File Inventory. 

(F) Per journal policy, if you have generated any custom code during the course of this investigation, please make it available without restrictions. Please ensure that the code is sufficiently well documented and reusable, and that your Data Statement in the Editorial Manager submission system accurately describes where your code can be found.

(G) Please ensure that your Data Statement in the submission system accurately describes where your data can be found and is in final format, as it will be published as written there. 

We expect to receive your revised manuscript within two weeks. 

*Published Peer Review History*

*Press*

Best wishes,

Richard

Richard Hodge, PhD

rhodge@plos.org

PLOS

---

## [Editor Report · Decision Letter 3]

6 Dec 2024

Dear Hiroki,

On behalf of my colleagues and the Academic Editor, Takeshi Tsubata, I am pleased to say that we can accept your manuscript for publication, provided you address any remaining formatting and reporting issues. These will be detailed in an email you should receive within 2-3 business days from our colleagues in the journal operations team; no action is required from you until then. Please note that we will not be able to formally accept your manuscript and schedule it for publication until you have completed any requested changes.

PRESS

Best wishes, 

Richard

Richard Hodge, PhD

rhodge@plos.org

PLOS
